# *MedDelinea*: Scalable and Efficient Medical Image Segmentation via Controllable Diffusion Transformers

**Gayatri Deshmukh**[†]                                          DGAYATRI9850@GMAIL.COM
**Onkar Kishor Susladkar**[†]                              ONKARSUS13@GMAIL.COM
**Debesh Jha**[*]                                                      DEBESH.JHA@USD.EDU
**Elif Keles**[†]                                          ELIF.KELES@NORTHWESTERN.EDU
**Ertugrul Aktas**[†]                                      H.ERTUGRULAKTAS@GMAIL.COM
**Alpay Medetalibeyoglu**[†]                          ALIBEYOGLUALPAY@GMAIL.COM
**Daniela P. Ladner**[†]                                            DLADNER@NM.ORG
**Amir A. Borhani**[†]                              AMIR.BORHANI@NORTHWESTERN.EDU
**Gorkem Durak**[†]                              GORKEM.DURAK@NORTHWESTERN.EDU
**Ulas Bagci**[†]                                    ULAS.BAGCI@NORTHWESTERN.EDU
*[†]Machine and Hybrid Intelligence Lab, Northwestern University, Chicago, IL*
*[*]University of South Dakota*

**Editors:** Accepted for publication at MIDL 2025

## Abstract

We introduce **MedDelinea**, a novel medical image segmentation architecture that leverages a controllable module, drawing inspiration from *ControlNet*, within the Diffusion Transformers (DiT) framework. By doing so, we effectively address three key challenges inherent to segmentation tasks: (1) limited availability of labeled data, (2) variability in image modalities, and (3) the need for precise boundary delineation. **MedDelinea** is pre-trained on a large-scale medical dataset, thereby mitigating overfitting risks and enabling efficient transfer across diverse imaging scenarios with minimal fine-tuning requirements. The modular design of **MedDelinea** facilitates scalable and efficient computation, while maintaining high-quality segmentation performance in both supervised and zero-shot settings. Through extensive empirical evaluations on multiple datasets, we demonstrate that **MedDelinea** outperforms existing state-of-the-art segmentation approaches, showcasing its potential for robust and accurate medical image analysis. The code is publicly available at: https://github.com/Onkarsus13/MedDelinea.

## 1. Introduction

Generalization is an unsolved problem in medical image segmentation, and various pre-training strategies have been explored to this challenge. Self-pretraining (Tang et al., 2022), where a model is pre-trained on the same dataset used for downstream tasks, helps the model adapt to the specific characteristics of medical images but risks overfitting, limiting generalization to new data. In contrast, pre-training on large, diverse datasets with transformer-based or CNN models improves generalization but requires extensive fine-tuning for specific medical tasks, reducing efficiency. This trade-off between generalization and efficiency hinders the practical use of these models. To address these limitations, we design a model with four key attributes: Transferability, Efficiency, Modularity, and Scalability. A transferable model requires minimal fine-tuning across related tasks. As shown in Fig. 1, **MedDelinea** demonstrates gradual loss reduction with fewer iterations compared to others. Efficiency focuses on high accuracy with minimal annotation. Modularity enables reuse across tasks without major modifications, while Scalability ensures performance remains stable as data size or complexity grows.

In this study, we introduce **MedDelinea**, a Large-Scale Supervised Pre-training model with Diffusion Transformer, aimed at achieving our primary goals in abdominal segmentation. **MedDelinea** features two key innovations: 1) Novel Architecture Design: Departing from the traditional UNet backbone, it incorporates elements from ControlNet, Latent Diffusion Models (LDMs), and Vision Transformers. This replaces UNet-style diffusion models with transformers and leverages pre-trained blocks from ControlNet to balance domain-agnostic and domain-specific learning, resulting in a modular, scalable, and efficient architecture. 2) Pretraining Strategy: By using a ControlNet-based framework and performing supervised pretraining on a large-scale dataset, **MedDelinea** enhances transferability and generalization, providing precise control over segmentation outputs to meet the demands of complex anatomical structures. Summary of our contributions are as follows:

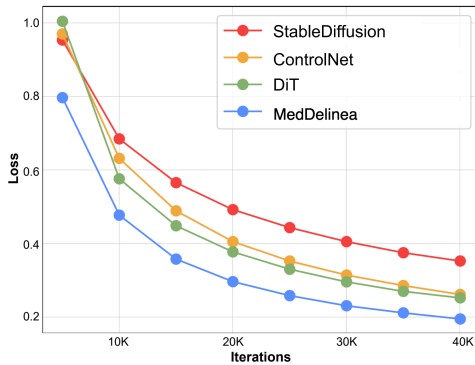

**Figure 1:** Learning curve of different models across iterations. MedDelinea consistently demonstrates the lowest loss through the training process.

- **Innovative Model Architecture**: We propose a novel architecture, **MedDelinea**, which integrates controllable neural networks inspired by *ControlNet* with *Diffusion Transformers (DiT)*. This hybrid model is specifically designed to address challenges in medical image segmentation such as generalization to other datasets and minimal fine-tuning requirements.

- **High Segmentation Accuracy with Zero-Shot and Few-Shot Learning**: **MedDelinea** leverages a large-scale dataset (ATLAS-8k) for pretraining, enabling enhanced transferability and zero-shot and few-shot segmentation capabilities. This approach allows the model to perform effectively on new datasets without additional fine-tuning. Extensive evaluations demonstrate that **MedDelinea** achieves state-of-the-art performance in segmenting complex anatomical structures with high precision, contributing to improved diagnostics and treatment planning in medical imaging.

- **Robustness, Modularity, and Scalability**: The architecture of **MedDelinea** is both modular and scalable, making it adaptable to diverse medical imaging tasks without significant computational overhead or architectural modifications. **MedDelinea** shows consistent and robust performance across various imaging modalities, including CT and MRI, and effectively handles multiple object segmentation within these modalities, highlighting its versatility and efficiency.

## 1.1. Related Work

**Emergence of Diffusion Models in Medical Image Segmentation:** Recently, diffusion models have emerged as a promising approach in medical image segmentation by framing the task as a generative modeling problem. Denoising Diffusion Probabilistic Models (DDPMs) (Ho et al., 2020) have been adapted for segmentation, where they treat it as conditional image generation. PD-DDPM (Guo et al., 2023) improves efficiency by leveraging pre-segmentation results, while BerDiff (Chen et al., 2023) uses a Conditional Bernoulli Diffusion Model for binary segmentation tasks. MedSegDiff (Wu et al., 2024) utilizes diffusion processes to handle uncertain cases, such as tumor boundaries. However, many of these models still rely on UNet backbones, limiting their ability to capture global features. MedSegDiff-V2 (Wu et al., 2024) overcomes this by integrating transformers with diffusion models, enhancing global feature capture and improving segmentation quality.

**Pre-Training in Medical Imaging:** Pre-training methods have become increasingly important in medical image segmentation, especially due to the scarcity of large annotated datasets. These approaches improve model performance by learning meaningful representations from large, unlabeled datasets. Self-supervised learning (SSL) is a prominent pre-training strategy that helps models learn from unlabeled data through proxy tasks. SimCLR (Ali et al., 2021), for example, applies contrastive learning to discover underlying data structures, while Swin-UNETR (Tang et al., 2022) uses proxy tasks like masked volume inpainting and rotation prediction to capture anatomical patterns in large CT datasets. Another model, UNetFormer (Hatamizadeh et al., 2022), reconstructs masked tokens to learn from visible input regions. These methods offer significant improvements in performance when labeled data is scarce. Task-specific self-pre-training has also gained attention. Zhou et al. (Zhou et al., 2023) utilize a Masked Autoencoder (MAE) for Vision Transformers (ViT), pre-training models by reconstructing masked images, enabling them to learn rich representations without extensive labeled datasets. This addresses a key challenge in medical imaging, where annotated data is often limited.

**Existing Gaps and Research Motivation:** Despite advances in transformer-based diffusion models like *MedSegDiff-V2*, challenges persist. Traditional reliance on UNet backbones limits these models in capturing global anatomical structures. SSL and self-pre-training methods show promise but overfitting to proxy tasks is a common problem, limiting generalization power of the networks, and making them less adaptable across tasks.

To address these gaps, we propose a novel hybrid architecture combining *Diffusion Transformers (DiT)* (Peebles and Xie, 2023) with *ControlNet* (Zhang et al., 2023). This approach integrates the generative strengths of diffusion models with transformers' ability to capture global features. Pre-training on large datasets followed by fine-tuning for specific segmentation tasks optimizes accuracy, complexity, and adaptability, offering a more scalable and efficient solution for medical image segmentation across diverse clinical applications.

## 2. Method

An overview of our proposed Controllable Diffusion Transformers (DiT) architecture for medical image segmentation is presented in Fig. 2. By leveraging the robust representations learned by pre-trained models on large-scale datasets, our approach enables efficient adaptation to the segmentation task with minimal fine-tuning or even zero-shot learning. The DiT framework is particularly well-suited for modeling complex data distributions inherent to medical images, while the integration of a control mechanism via *ControlNet* allows for seamless incorporation of external conditioning data to inform the segmentation process

**Pre-training:** Our pre-training strategy for ***MedDelinea*** involves leveraging the diverse anatomical coverage of the ATLAS-8k dataset (Qu et al., 2024) to learn robust features for medical image segmentation. We initialize the Control DiT Module and Pre-trained DiT Module weights from a pre-trained model, as described in (Peebles and Xie, 2023). During pre-training, we adopt a selective training approach, where only the Control DiT Module and the final layer of the VAE Decoder are updated, while the remaining model weights are frozen to preserve their generalization capabilities. This strategy is also employed during fine-tuning on other medical datasets, with the key distinction being that the Control DiT Module and Decoder weights are initialized from the pre-trained ***MedDelinea*** model, rather than directly from (Peebles and Xie, 2023). By doing so, we facilitate efficient transfer learning while maintaining the model's ability to generalize. A detailed description of each module is provided in the subsequent sections. Additionally, further pre-training details are included in the supplementary section.

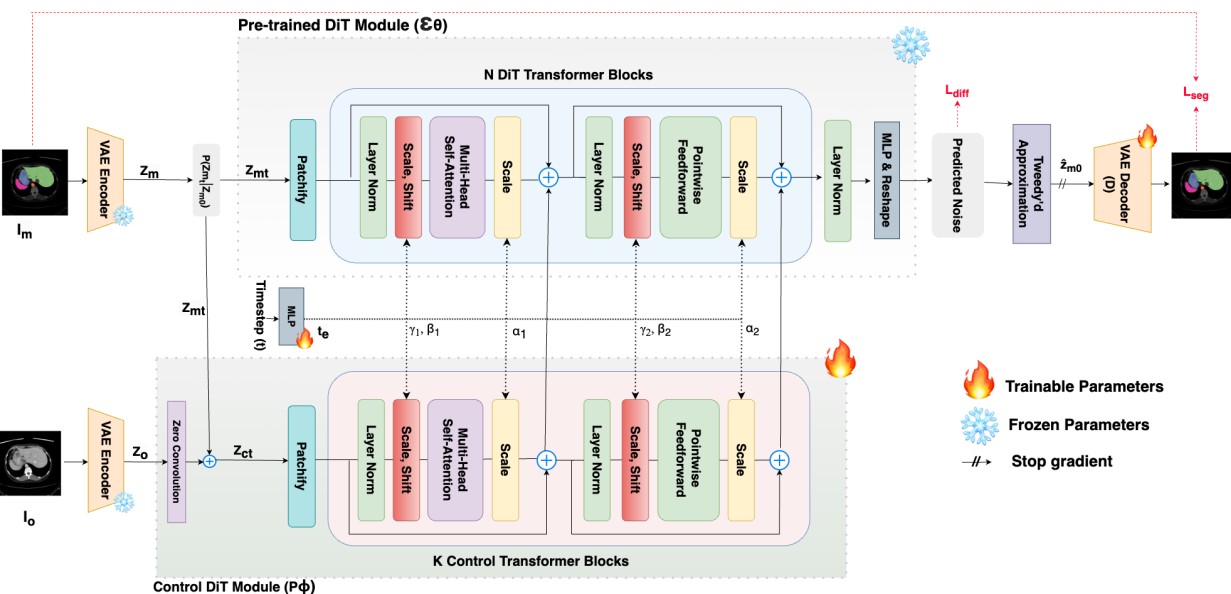

**Figure 2:** Architecture Diagram of **MedDelinea**. $\alpha_1$, $\alpha_2$, $\gamma_1$, $\gamma_2$, $\beta_1$ and $\beta_2$ represents scale and shift parameters.

**Input Processing and Latent Space Transformation:** Our model processes two input streams: a segmentation mask $(I_M)$ and an organ image $(I_O)$. These inputs are encoded into their respective latent representations, $Z_m$ and $Z_o$, using a pre-trained Variational Autoencoder (VAE) (Kingma, 2013) with frozen weights. By fixing the VAE's parameters, thanks to its ability to learn compact and informative latent representations, during this encoding step, we ensure that the resulting latent representations remain consistent and invariant, which in turn reduces the risk of overfitting and promotes stable performance across diverse input data.

The latent vector $Z_o$ corresponding to the CT/MRI image is further processed through a zero convolution layer, where the weights and biases are initialized to zero. This operation ensures that the initial influence of $Z_o$ on the subsequent diffusion process is neutral, allowing the model to adaptively learn the most relevant features from the condition image as the training progresses. On the other hand, $Z_m$ undergoes a forward diffusion process, resulting in a noisy latent vector $Z_{mt}$. $P(Z_{mt}|Z_{m0}) = \sqrt{\bar{\alpha}_t}Z_{m0} + \sqrt{1 - \bar{\alpha}_t}\epsilon_t$, where $\epsilon_t \sim \mathcal{N}(0,1)$. This noisy representation $Z_{mt}$ is then concatenated with $Z_o$, producing a conditional latent vector $Z_{ct}$. This combination allows the model to embed the condition-specific information into the diffusion process, effectively guiding the generation of segmentation masks.

**Control DiT Module $(P_\phi)$:** Durining fine-tuining, the weights are initialized from the Control DiT Module pre-trained on ATLAS-8k dataset. The Control DiT Module is designed to manipulate the diffusion process by injecting conditional information derived from $Z_{ct}$. This module first converts $Z_{ct}$ into a sequence of tokens using a patchification process, where the input is divided into patches, each linearly embedded into a token of dimension $d$. This transformation allows the model to process the spatial information within the input in a manner compatible with transformer architectures.

As illustrated in Fig. 2, the tokens are then processed through a sequence of $K$ transformer blocks. Each block incorporates a timestep embedding $t_e$, which serves as a conditioning signal for the scale and shift module within the block. This enables the model to adapt its operations

**Table 1:** Mean segmentation metrics for the ATLAS-8k dataset.

| Methods | mDSC ($\uparrow$) | mHD95 ($\downarrow$) | mASSD ($\downarrow$) |
|---|---|---|---|
| UNet (Ronneberger et al., 2015) | 75.32 | 33.42 | 14.32 |
| TransUnet (Chen et al., 2021) | 78.92 | 31.78 | 12.11 |
| SynergyNet (Gorade et al., 2024) | 80.32 | 30.28 | 10.43 |
| ControlNet (Zhang et al., 2023) | 83.77 | 28.73 | 10.01 |
| MedSegDiff (Wu et al., 2024) | 85.07 | 27.99 | 9.27 |
| VQDiffusion (Gu et al., 2022) | 84.92 | 28.01 | 9.33 |
| DiT (Peebles and Xie, 2023) | 85.48 | 28.24 | 9.01 |
| *MedDelinea* | **87.98** | **25.92** | **7.98** |

dynamically according to the diffusion stage, thereby maintaining coherence in the generated images throughout the diffusion process. Furthermore, the Control DiT Module interacts with the Pre-trained DiT Module by injecting feature vectors at each residual connection point of the DiT transformer blocks. These feature vectors act as external guides, providing conditional information that informs the denoising operations within the pre-trained DiT transformer blocks. Notably, the parameters of the Control DiT Module remain trainable, allowing the model to learn task-specific adaptations and refine its performance on the target segmentation task.

**Pre-trained DiT Module ($\epsilon_\theta$):** The Pre-trained DiT Module plays a critical role in refining the noisy latent representation $Z_{mt}$. This module, which consists of $N$ DiT transformer blocks, is entirely frozen during training. By freezing these blocks, the architecture preserves the pre-learned knowledge from the extensive training on large datasets (Peebles and Xie, 2023), ensuring that the model retains its ability to generalize while focusing on the new dataset.

Similar to the Control DiT Module, the Pre-trained DiT Module begins by patchifying the input $Z_{mt}$ into a sequence of tokens. These tokens are then processed by the frozen transformer blocks. The scaling and shifting within these blocks are guided by the feature vectors passed from the Control DiT Module. This design allows the model to fine-tune the diffusion process based on the condition-specific information, ensuring that the denoising operation aligns with the desired segmentation outcome. After processing through the transformer blocks, the latent vector is approximated to its initial state $Z_{m0}$ using Tweedie's formula (Efron, 2011), which is a well-established method in denoising processes for estimating the clean latent from noisy distribution. The resultant vector is then passed to the VAE Decoder.

$$\hat{Z_{m0}}(Z_{mt}) = \frac{Z_{mt} - \sqrt{1 - \bar{\alpha}_t} \cdot \epsilon_\theta(Z_{mt}, t, P_\phi(Z_{mt}, Z_i, t))}{\sqrt{\bar{\alpha}_t}}, \tag{1}$$

where, $\epsilon_\theta$ refers to the Pre-trained DiT module and $P_\phi$ refers to the Control DiT module.

**Image Reconstruction and Loss Functions:** The VAE Decoder, which remains trainable, reconstructs the final image from the refined latent space. The trainability of the VAE Decoder is crucial as it allows the model to adapt to the specific characteristics of the dataset, ensuring that the segmentation masks generated are both accurate and relevant to the dataset.

Two primary loss functions are employed to guide the training process:

*a) Diffusion Loss ($L_{diff}$):* This loss function is applied at the end of the Pre-trained DiT Module. It is designed to predict the noise present in the latent noisy vector $Z_{mt}$. The diffusion loss is typically defined as the mean squared error (MSE) between the predicted and actual noise, encouraging the model to accurately denoise the latent representation at each timestep: $L_{diff} = |\epsilon_t - \epsilon_\theta(Z_{mt}, t, P_\phi(Z_{mt}, Z_i, t))|^2$.

*b) Segmentation Loss ($L_{seg}$):* This loss function is applied to the final output image to ensure that the generated segmentation masks not only look visually plausible but also precisely delineate the anatomical structures of interest. This is particularly critical in medical imaging, where segmentation accuracy is paramount for diagnostic and treatment planning purposes: $L_{seg} = L_{BCE}\left(D(\hat{Z}_{m0}), I_m\right) + L_{Dice}\left(D(\hat{Z}_{m0}), I_m\right)$, where, $D$ is the VAE Decoder.

**Table 2:** Quantitative results on BTCV, AMOS, Cirr600+, and PanSegData datasets.

| Methods | BTCV | | AMOS | | Cirr600+ | | PanSegData | |
|---|---|---|---|---|---|---|---|---|
| | mDSC (↑) | mHD95 (↓) | mDSC (↑) | mHD95 (↓) | Dice (↑) | HD95 (↓) | Dice (↑) | HD95 (↓) |
| Models fine-tuned after ATLAS-8k pre-training | | | | | | | | |
| UNet | 70.88 | 30.98 | 69.01 | 33.21 | 56.78 | 39.77 | 54.18 | 31.45 |
| TransUnet | 73.47 | 28.77 | 71.65 | 31.72 | 60.11 | 36.83 | 61.92 | 33.93 |
| SynergyNet | 79.65 | 23.29 | 77.67 | 31.28 | 64.76 | 33.47 | 67.32 | 31.77 |
| ControlNet | 82.96 | 21.15 | 78.19 | 30.76 | 66.84 | 31.55 | 69.92 | 29.98 |
| MedSegDiff | 83.55 | 20.19 | 80.77 | 28.75 | 78.92 | 29.57 | 70.56 | 29.12 |
| VQDiffusion | 82.71 | 24.11 | 80.13 | 28.82 | 77.88 | 31.07 | 70.12 | 29.91 |
| DiT | 83.41 | 23.21 | 81.45 | 27.17 | 78.18 | 30.07 | 70.13 | 29.91 |
| *MedDelinea* | **87.01** | **19.00** | **82.19** | **26.56** | **87.63** | **26.95** | **72.03** | **28.01** |
| Zero-shot learning: Direct inference on test sets after ATLAS-8k pre-training | | | | | | | | |
| UNet | 59.76 | 36.67 | 54.57 | 39.21 | 47.82 | 38.72 | 49.12 | 33.47 |
| TransUnet | 61.19 | 34.22 | 60.01 | 36.54 | 48.77 | 37.14 | 52.77 | 32.88 |
| SynergyNet | 63.44 | 32.18 | 63.47 | 35.52 | 61.21 | 35.19 | 58.16 | 31.09 |
| ControlNet | 67.52 | 30.88 | 67.29 | 34.78 | 62.32 | 34.28 | 66.92 | 30.22 |
| MedSegDiff | 73.27 | 28.92 | 70.64 | 33.01 | 64.12 | 33.01 | 68.77 | 29.88 |
| VQDiffusion | 77.54 | 29.01 | 69.64 | 33.42 | 63.92 | 33.34 | 68.17 | 29.92 |
| DiT | 77.79 | 26.81 | 76.47 | 31.42 | 64.01 | 33.19 | 68.17 | 28.52 |
| *MedDelinea* | **80.01** | **23.11** | **78.67** | **29.51** | **65.17** | **30.12** | **70.52** | **27.81** |

By combining these loss functions the total loss on which model trained on is: $L_t = L_{diff} + L_{seg}$. The model is trained to produce high-quality segmentation masks that meet both aesthetic and clinical standards. The architecture's design, which integrates pre-trained knowledge with adaptive control mechanisms, ensures that the model is both robust and flexible, capable of performing complex segmentation tasks with minimal additional training.

## 3. Experiments and Results

Details regarding implementation details (Section A), along with information about the dataset and evaluation metrics (Section B), is provided in the Appendix.

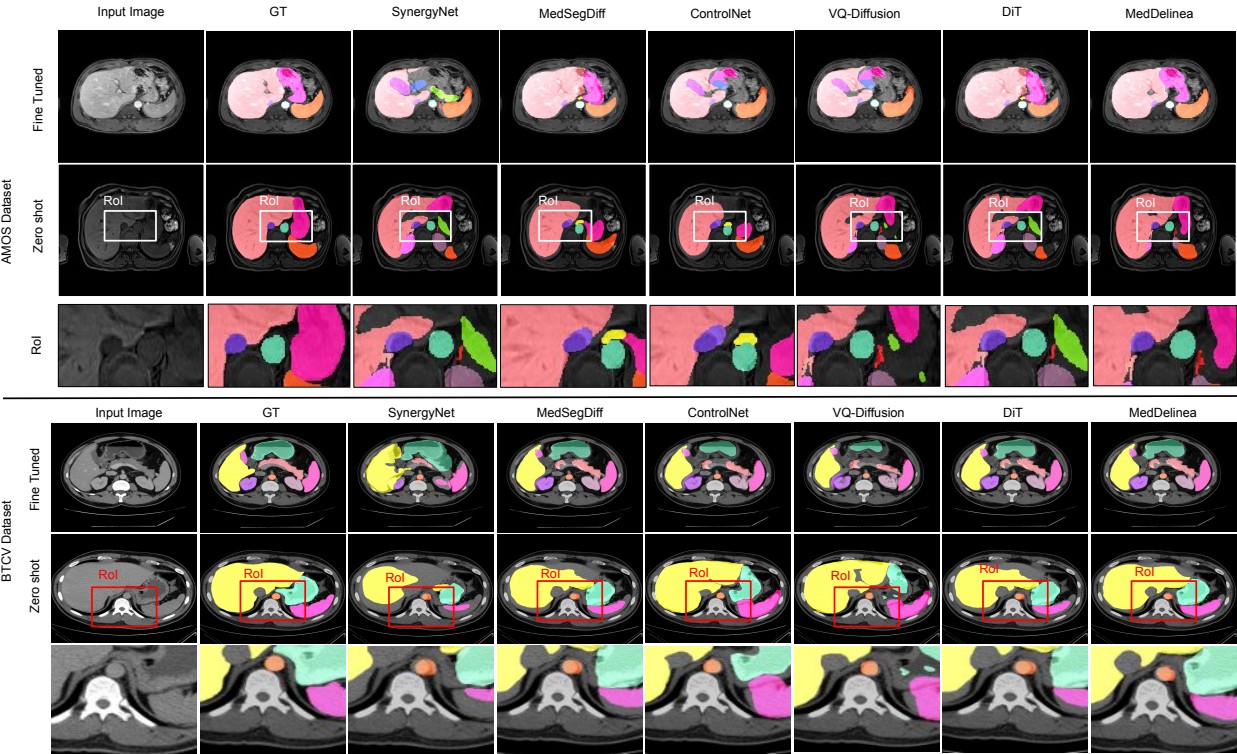

**Figure 3:** segmentation performance of various models on AMOS and BTCV datasets under fine-tuned and zero-shot settings, highlighting the region of interest (RoI) for detailed anatomical segmentation analysis.

**Table 3:** Few-shot learning results: Models fine-tuned on 1%, 10%, and 25% of training data after ATLAS-8k pre-training, followed by inference on test sets.

| Methods | BTCV | | AMOS | |
|---|---|---|---|---|
| | mDSC (↑) | mHD95 (↓) | mDSC (↑) | mHD95 (↓) |
| *Results on 1% of Training Data* | | | | |
| ControlNet | 72.12 | 27.52 | 69.03 | 34.02 |
| DiT | 74.22 | 25.11 | 76.67 | 32.18 |
| *MedDelinea* | **77.52** | **23.42** | **78.92** | **29.92** |
| *Results on 10% of Training Data* | | | | |
| ControlNet | 72.82 | 27.03 | 71.28 | 33.05 |
| DiT | 74.97 | 24.55 | 77.01 | 31.51 |
| *MedDelinea* | **78.59** | **22.78** | **79.23** | **28.79** |
| *Results on 25% of Training Data* | | | | |
| ControlNet | 74.02 | 26.38 | 73.34 | 31.27 |
| DiT | 75.11 | 24.01 | 79.34 | 29.16 |
| *MedDelinea* | **81.16** | **21.59** | **80.46** | **27.49** |

## 3.1. Quantitative Comparison

**Fine-tuining and zero-shot learning Scenarios:** Table 2 presents the quantitative analysis of **MedDelinea**'s performance in fine-tuning and zero-shot learning scenarios. After fine-tuning on each dataset post ATLAS-8k pre-training, **MedDelinea** achieves the highest metrics, including an mDSC of 87.01% on BTCV and 82.19% on AMOS. In zero-shot learning, **MedDelinea** demonstrates robust generalization capabilities, leading with an mDSC of 80.01% on BTCV and 78.67% on AMOS. This suggests that **MedDelinea**'s architecture is adaptable and effective in transferring learned features from pre-training to new datasets.

The combination of ControlNet and Diffusion Transformers in **MedDelinea** enables it to generalize complex features and excel in zero-shot learning. In contrast, other models like MedSegDiff and VQDiffusion, while competitive in fine-tuning, struggle with boundary accuracy and surface accuracy in zero-shot scenarios, as indicated by higher mHD95 and mASSD values on challenging datasets.

**Few-shot Comparison:** Table 3 presents the few-shot learning performance of different models, including **MedDelinea**, fine-tuned on 1%, 10%, and 25% of the training data after ATLAS-8k pre-training. **MedDelinea** consistently outperforms other models (ControlNet and DiT) across varying amounts of training data, achieving the highest mDice scores and lowest mHD95 values on both BTCV and AMOS datasets. Note that, with only 1% of the training data, **MedDelinea** achieves mDice scores of 77.52% (BTCV) and 78.92% (AMOS), demonstrating its robustness and efficiency in few-shot learning scenarios. As the training data increases, **MedDelinea** further solidifies its lead, showcasing its superior scalability and effectiveness.

**Altas8k Dataset:** Table 1 shows that our proposed method, **MedDelinea**, outperforms previous baselines with superior scalability. The incorporation of the Control DiT Module in **MedDelinea** mitigates the issue of weight distribution shift caused by learning new segmentation tasks, leading to improved scalability and adaptability. This confirms that **MedDelinea** is a more reliable and efficient model for segmentation tasks. Organ-specific Dice scores for ATLAS-8K are provided in Appendix Table 9.

Details on the zero-shot segmentation performance analysis of models on MRI and CT, are provided in the supplementary section.

**Inference Time and Model Parameter Comparison:** A detailed comparison of the number of parameters and inference time across baseline models and **MedDelinea** is provided in the Table 4. To ensure a fair comparison, all inference evaluations were conducted on a single NVIDIA A6000 GPU. **MedDelinea** consists of 227 million parameters and achieves state-of-the-art (SOTA) performance with competitive inference time relative to other models. While MedDelinea has a slightly larger parameter count compared to TransUNet and DiT, it maintains efficient inference time while significantly outperforming these models in segmentation accuracy. Notably, it is less

**Table 4:** Comparison of model parameters (in millions) and inference time (in seconds) across various baseline methods and MedDelinea.

| Methods | Parameters M | Inference Time |
|---|---|---|
| UNet | 33 | 0.25 |
| TransUnet | 182 | 0.76 |
| SynergyNet | 23 | 0.46 |
| ControlNet | 400 | 6.32 |
| MedSegDiff | 372 | 8.89 |
| VQDiffusion | 340 | 3.11 |
| DiT | 187 | 2.72 |
| MedDelinea | 227 | 2.88 |

**Table 5:** Ablation Study. Default settings indicate the architectural choices used for all experiments.

| Experiments | mDSE (↑) | mHD95 (↓) | mASSD (↓) |
|---|---|---|---|
| Ablation Based on Training Stratergy | | | |
| Training only Noise Predictor | 86.21 | 26.11 | 08.03 |
| Abaltion Based on loss (Default: $L_{diff}$ and $L_{seg}$) | | | |
| Only $L_{diff}$ | 86.92 | 26.01 | 08.01 |
| Ablation based on Number of Control-DiT Transformer Blocks (Default: K=15) | | | |
| K=1 | 86.57 | 26.54 | 08.33 |
| K=5 | 86.72 | 26.37 | 08.27 |
| K=10 | 87.37 | 25.98 | 08.01 |
| Abaltion Based on Noise Schedulers (Default: EulerAncestralDiscreteScheduler (Karras et al., 2022)) | | | |
| DDIM (Song et al., 2020) | 87.51 | 25.32 | 08.11 |
| DPMSolver++ (Lu et al., 2022) | 87.22 | 25.45 | 08.11 |
| UniPCMultistepScheduler (Zhao et al., 2024) | 87.92 | 25.78 | 07.97 |
| LMSDiscreteScheduler (Karras et al., 2022) | 87.76 | 25.34 | 07.96 |

computationally expensive than MedSegDiff, VQ-Diffusion, and ControlNet while delivering superior performance. This comparison underscores ***MedDelinea's*** ability to balance model complexity and inference efficiency, making it a highly effective solution for medical image segmentation.

## 3.2. Qualitative Comparison

Qualitative results in Fig. 3 demonstrate ***MedDelinea***'s superior segmentation performance on BTCV and AMOS datasets in both fine-tuned and zero-shot scenarios. Compared to other models (e.g., MedSegDiff, ControlNet, VQ-Diffusion, and DiT), ***MedDelinea*** consistently delivers accurate segmentations with fewer artifacts and mis-segmentations. Notably, ***MedDelinea*** excels in delineating organ boundaries, particularly in the pancreas and spleen (BTCV) and liver, kidneys, and pancreas (AMOS). MedDelinea captures fine details and complex structures, achieving superior boundary delineation and segmentation performance. See Appendix Fig. 6 for attention maps.

Additional quantitative class-wise metrics for the BTCV and AMOS datasets (See Tables 5 and 6 in Appendix), along with qualitative results from ATLAS-8k (See Fig. 5 in Appendix) and 3D segmentation results for BTCV and AMOS (See Fig. 7 in Appendix), are provided in the appendix.

## 3.3. Ablation Study

We conducted ablation studies to assess the impact of different components and configurations on the model's performance, using the Atlas-8K dataset for these experiments. The results are summarized in Table 5. All results for the default choices are presented in Table 1.

**Ablation Based on Training Stratergy:** We analyzed the impact of different training strategies for the decoder and noise predictor. Training only the noise predictor yielded lower performance, with an mDSE of 86.21, mHD95 of 26.11, and mASSD of 8.03. However, training both the noise predictor and the decoder together (default choice) improved results across all metrics (refer Table 1).

**Ablation Based on Loss:** Using only diffusion loss ($L_{diff}$) resulted in poor boundary accuracy. Adding segmentation loss ($L_{seg}$) with a BCE Dice component improved performance, raising

the mDSE from 86.92 to 87.98, lowering mHD95 from 26.01 to 25.92, and improving mASSD from 8.01 to 7.98. This highlights the value of segmentation-specific losses for better boundary accuracy.

**Ablation based on Number of Control-DiT Transformer Blocks:** Increasing the number of Control-DiT transformer blocks (K) led to better segmentation performance. With K rising from 1 to 15, the model reached its best scores—mDSE of 87.98, mHD95 of 25.92, and mASSD of 7.98. However, this improvement came with a substantial increase in trainable parameters, from 7.83 million at K=1 to 117.45 million at K=15.

**Ablation Based on Noise Schedulers:** The results in Table 1 show that the Euler Ancestral Discrete Scheduler outperforms others, with the highest mDSE score (87.98), fastest inference time (2.78s), and strong boundary accuracy. In comparison, the UniPC and LMSDiscreteSchedulers (Table 5) have slightly lower mDSE scores (87.92 and 87.76) and longer inference times (8.54s and 5.32s). This indicates the Euler Ancestral Discrete Scheduler offers the best balance of speed and accuracy.

## 4. Conclusion

In conclusion, **MedDelinea** presents a novel and scalable architecture that significantly enhances medical image segmentation by integrating controllable neural networks with Diffusion Transformers (DiT). The model demonstrates strong generalization capabilities, achieving state-of-the-art performance with minimal fine-tuning and excelling in zero-shot settings. Its modular design allows for efficient adaptation across various medical imaging tasks, ensuring both computational efficiency and high accuracy. **MedDelinea**'s success highlights the potential of leveraging pre-trained models combined with adaptive control mechanisms, offering a robust solution for diverse and complex medical image segmentation challenges.

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

## Appendix A. Training and Implementation Details

**Pre-training Details:** The *MedDelinea* model was pre-trained over 35,000 steps using the AdamW optimizer with a learning rate of 2.0e-5. A Cosine annealing scheduler was applied for learning rate decay. The model was trained on 8 Nvidia A6000 GPUs (48 GB each), with a batch size of 32 per GPU. Image resolutions were dynamically adjusted per batch, alternating between $256 \times 256$ and $512 \times 512$. Gradient accumulation was set to 8 steps, and mixed precision training was employed to improve computational efficiency and reduce memory usage. Pre-training was conducted using the Atlas 8K dataset, following a 90:10 split for training and validation.

**Fine-tuning Details:** Fine-tuning was conducted for 1,500 steps using the same AdamW optimizer with a learning rate of 2.0e-5. A Linear scheduler was used for learning rate adjustments. The batch size remained at 32 per GPU, with the same hardware configuration. During fine-tuning, the image resolution was fixed at $256 \times 256$. Similar to pre-training, gradient accumulation was set to 8 steps, and mixed precision training was used to optimize both speed and memory efficiency. This fine-tuning process was consistently applied across all datasets. Fine-tuning and evaluation were performed using the BTCV, AMOS, and CirrMri600+ datasets, following a standardized 80:10:10 split for training, validation, and testing.

**Data Augmentation:** To improve generalization and robustness to variations in medical imaging data, we applied the same data augmentation strategies during both pre-training and fine-tuning across *MedDelinea* and all baseline methods. The augmentation techniques included Random Cropping, which extracts random patches to introduce spatial variability, and Random Affine Transformations, which apply scaling, shearing, and translation to enhance invariance to spatial deformations. Additionally, Elastic Distortion was employed to simulate realistic non-rigid deformations commonly seen in medical scans, while Random Rotation introduced angular perturbations to improve robustness to orientation differences. These augmentations were consistently applied throughout both training stages, ensuring that the learned representations remain invariant to common geometric transformations and distortions in medical imaging data.

## Appendix B. Datasets and Metrics

**Datasets:** To evaluate *MedDelinea*'s performance in medical image segmentation, we fine-tuned the model on three datasets after pre-training on ATLAS-8k (Qu et al., 2024), which contains 8,000 CT scans. For fine-tuning, we used BTCV (Fang and Yan, 2020), AMOS (Ji et al., 2022), Cirr600+ (Jha et al., 2024), and PanSegData (Zhang et al., 2024). BTCV includes 50 multi-organ abdominal CT scans for benchmarking. AMOS provides 600 volumes (500 CT, 100 MRI) for multi-modal segmentation of abdominal organs. Cirr600+ features 608 MRI scans focused on cirrhotic liver cases, and Pancreas-MRI comprises 767 MRI scans from 499 participants across five centers. This diverse dataset combination enhances *MedDelinea*'s robustness and generalizability across medical imaging scenarios.

**Metrics:** To assess model performance, we used mean Dice Similarity Coefficient (mDSC), mean Hausdorff Distance 95 (mHD95), and Dice. mDSC and Dice quantify segmentation overlap, where 1 indicates perfect agreement. mHD95 measure boundary errors and surface distances, with lower values indicating better accuracy and precision. These metrics offer a comprehensive evaluation of segmentation accuracy and boundary precision.

## Appendix C. Detailed Pre-training Strategy

Our pre-training strategy leverages the ATLAS-8K dataset, which contains 8,000 3D CT volumes annotated with 15 abdominal structures. The dataset provides comprehensive anatomical coverage, allowing models to learn robust feature representations for medical image segmentation. During

pre-training, we initialize the Control DiT Module and Pre-trained DiT Module weights from a model pre-trained as described in (Peebles and Xie, 2023). However, we employ selective training, wherein only the Control DiT Module and the final layer of the VAE Decoder are updated, while the remaining weights remain frozen. This preserves the generalization capability of the original model while allowing for effective adaptation to medical imaging.

**Motivation for Pre-training:** While popular diffusion models such as DiT, Stable Diffusion, and VQ-Diffusion are pre-trained on natural images, they lack domain-specific knowledge crucial for medical image segmentation. Directly applying them to CT/MRI data leads to suboptimal generalization due to differences in contrast, texture, and domain-specific noise characteristics. Our pre-training on ATLAS-8K addresses this issue in the following ways:

- Domain Alignment: The dataset's intensity distributions, anatomical priors, and noise characteristics ensure the model learns medical image representations rather than natural image statistics.

- Improved Generalization: Exposure to diverse anatomical variations improves the model's ability to segment structures across different CT/MRI scans.

- Segmentation-Specific Guidance: We integrate segmentation loss at the decoder stage, reinforcing feature learning for anatomical edges and overlapping structures.

## Appendix D. Quantitative Results

Tables 6 and 7 present the classwise Dice scores for various models on the BTCV (Fang and Yan, 2020) and AMOS (Ji et al., 2022) datasets, highlighting both fine-tuned and zero-shot results after pre-training on the ATLAS-8k (Qu et al., 2024) dataset.

In Table 6, the models were fine-tuned on the BTCV and AMOS datasets. ***MedDelinea*** consistently achieves the highest Dice scores across nearly all organs in both datasets. On the BTCV dataset, ***MedDelinea*** shows superior performance in complex organs such as the liver (96.72%) and pancreas (63.77%), outperforming models like MedSegDiff and DiT, which also perform well but not to the same degree of accuracy. Similarly, in the AMOS dataset, ***MedDelinea*** achieves the highest Dice scores for the liver (93.14%) and spleen (91.72%), maintaining its edge across most organ classes.

Table 7 presents the results from zero-shot inference on the BTCV and AMOS test sets without additional fine-tuning. Again, ***MedDelinea*** outperforms other models, particularly in difficult

**Table 6:** Classwise quantitative results on the BTCV and AMOS datasets, where models are fine-tuned on these datasets after pre-training on the ATLAS-8k dataset.

| Methods | Dice (↑) | | | | | | | |
|---|---|---|---|---|---|---|---|---|
| | Aorta | Gallbladder | KidneyL | KidneyR | Liver | Pancreas | Spleen | Stomach |
| *BTCV* | | | | | | | | |
| UNet | 75.12 | 59.77 | 72.31 | 70.88 | 86.67 | 47.78 | 80.19 | 73.21 |
| TransUnet | 77.05 | 61.92 | 74.11 | 73.08 | 88.79 | 49.97 | 81.37 | 73.98 |
| SynergyNet | 86.10 | 65.49 | 82.78 | 78.24 | 95.06 | 58.28 | 88.95 | 81.30 |
| ControlNet | 89.19 | 68.78 | 85.92 | 83.19 | 95.98 | 61.09 | 90.28 | 85.09 |
| MedSegDiff | 89.92 | 69.03 | 86.67 | 83.32 | 96.88 | 61.12 | 90.11 | 84.11 |
| VQDiffusion | 88.56 | 66.77 | 85.12 | 83.23 | 94.11 | 59.92 | 88.63 | 81.01 |
| DiT | 88.88 | 67.42 | 86.15 | 84.11 | 95.02 | 61.21 | 88.91 | 84.21 |
| *MedDelinea* | **91.77** | **72.78** | **88.09** | **85.18** | **96.72** | **63.77** | **92.96** | **86.61** |
| *AMOS* | | | | | | | | |
| UNet | 74.01 | 55.71 | 71.88 | 67.82 | 86.02 | 42.21 | 77.88 | 71.18 |
| TransUnet | 75.82 | 56.86 | 73.42 | 69.01 | 87.18 | 44.78 | 81.82 | 72.01 |
| SynergyNet | 83.89 | 63.16 | 81.01 | 82.11 | 88.12 | 44.17 | 87.19 | 84.12 |
| ControlNet | 84.99 | 64.77 | 82.76 | 82.62 | 89.19 | 45.22 | 89.02 | 85.32 |
| MedSegDiff | 86.08 | 65.92 | 83.44 | 83.19 | 91.78 | 46.04 | 90.65 | 86.78 |
| VQDiffusion | 85.19 | 65.01 | 84.17 | 83.02 | 90.72 | 45.99 | 89.77 | 85.27 |
| DiT | 85.99 | 64.88 | 84.76 | 82.92 | 91.44 | 45.34 | 90.02 | 84.98 |
| *MedDelinea* | **87.12** | **66.94** | **87.12** | **84.05** | **93.14** | **48.19** | **91.72** | **87.67** |

**Table 7:** Classwise quantitative results on the BTCV and AMOS datasets, obtained through zero-shot inference on the test set following fine-tuning on the ATLAS-8k dataset.

| Methods | Dice (↑) | | | | | | | |
|---|---|---|---|---|---|---|---|---|
| | Aorta | Gallbladder | KidneyL | KidneyR | Liver | Pancreas | Spleen | Stomach |
| *BTCV* | | | | | | | | |
| UNet | 69.67 | 54.12 | 65.43 | 65.24 | 80.11 | 39.12 | 71.21 | 59.77 |
| TransUnet | 71.22 | 55.77 | 69.88 | 66.54 | 82.11 | 40.14 | 73.22 | 61.01 |
| SynergyNet | 83.01 | 58.92 | 74.57 | 72.34 | 86.78 | 49.92 | 78.56 | 70.99 |
| ControlNet | 84.33 | 61.72 | 78.22 | 76.52 | 88.99 | 53.17 | 79.90 | 73.42 |
| MedSegDiff | 85.19 | 62.23 | 80.01 | 78.12 | 90.11 | 51.02 | 81.27 | 76.77 |
| VQDiffusion | 83.19 | 59.12 | 79.11 | 79.88 | 90.02 | 54.92 | 84.22 | 78.71 |
| DiT | 85.33 | 60.61 | 81.21 | 80.53 | 87.15 | 57.33 | 86.17 | 80.11 |
| *MedDelinea* | **89.17** | **68.18** | **85.19** | **83.09** | **94.21** | **60.88** | **89.78** | **83.10** |
| *AMOS* | | | | | | | | |
| UNet | 63.11 | 46.44 | 63.17 | 54.72 | 68.34 | 27.55 | 60.75 | 60.53 |
| TransUnet | 65.12 | 48.34 | 64.77 | 59.11 | 70.57 | 31.51 | 65.32 | 63.21 |
| SynergyNet | 68.92 | 50.02 | 69.24 | 71.02 | 73.76 | 36.28 | 71.54 | 66.54 |
| ControlNet | 71.12 | 56.72 | 73.77 | 74.57 | 78.91 | 38.91 | 77.33 | 74.58 |
| MedSegDiff | 70.92 | 58.21 | 75.26 | 75.72 | 82.65 | 40.42 | 82.37 | 79.91 |
| VQDiffusion | 71.24 | 54.11 | 77.56 | 77.01 | 83.72 | 38.76 | 80.22 | 77.82 |
| DiT | 75.12 | 60.21 | 81.56 | 77.01 | 87.46 | 41.42 | 80.22 | 77.82 |
| *MedDelinea* | **78.44** | **62.01** | **80.92** | **81.11** | **90.51** | **46.01** | **88.52** | **84.01** |

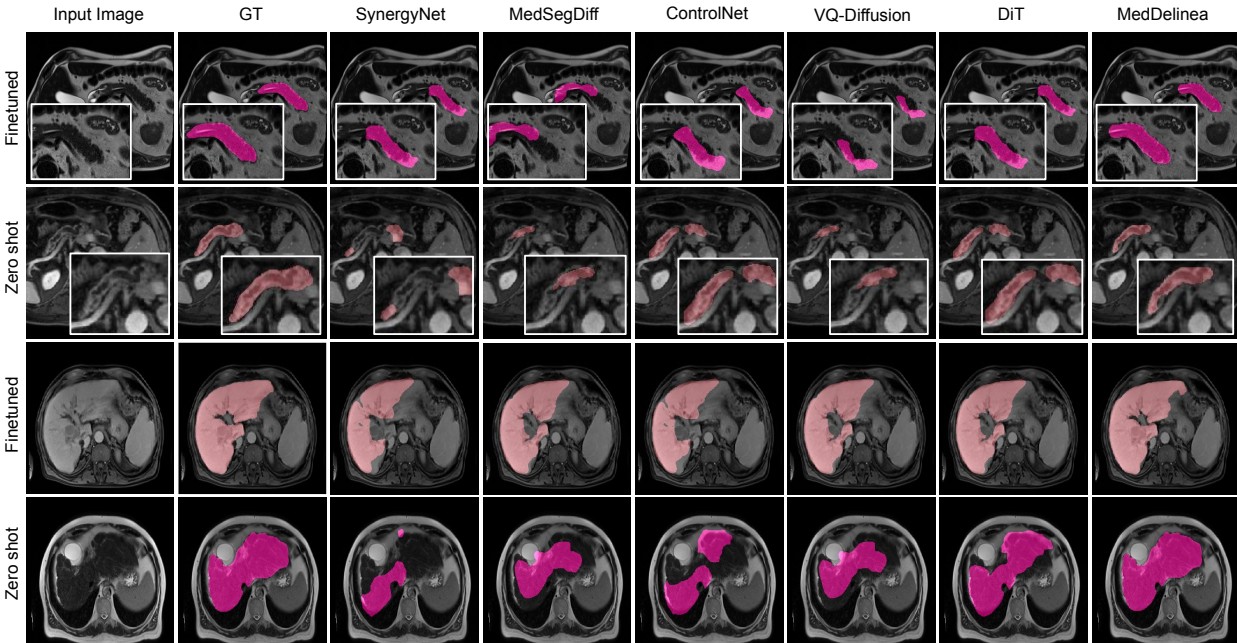

**Figure 4:** Pancreas and Liver dataset results.

organs like the liver (94.21% in BTCV, 90.51% in AMOS) and pancreas (60.88% in BTCV, 46.01% in AMOS) The reason for **MedDelinea**'s strong performance is its architecture choice which combines diffusion transformers (DiT) with a controllable module. This combination allows the model to effectively capture intricate anatomical structures. The use of extensive pre-training on the ATLAS-8k dataset helps **MedDelinea** build a strong representation of organ structures, making it more adept at handling fine-tuning on target datasets like BTCV and AMOS. This pre-training also explains its strong zero-shot performance, as it generalizes well to new organs without requiring further training.

Additionally, Table 9 reports the per-organ Dice scores (%) on the ATLAS-8k dataset. It is clear that our proposed method, MedDelinea, outperforms the other approaches across multiple organs, achieving higher Dice scores in most categories

**Zero-Shot Performance Analysis on MRI and CT:** To further evaluate the generalization capability of models pretrained solely on CT scans, we report the zero-shot segmentation performance on the AMOS MRI and CT subsets. All models were pretrained on ATLAS-8K (CT-only dataset) and tested on AMOS without any fine-tuning. The quantitative results are presented in Table 8. While the pretraining dataset (ATLAS-8K) comprises only CT scans, we observe that models maintain relatively high segmentation performance on MRI scans under zero-shot settings. However, as evident from Table 8, the performance on MRI remains consistently lower than on CT across all models; but still its competitive. This can be attributed to several factors:

1. Structural vs. Intensity-Based Representations in Diffusion Models: Our latent diffusion-based model, MedDelinea, prioritizes shape, spatial structure, and anatomical boundaries rather than absolute intensity values. Unlike CNN-based models, which may rely heavily on modality-specific intensity statistics, diffusion models focus on structural regularities, which are largely consistent across MRI and CT. This enables the model to achieve relatively robust MRI segmentation, despite having been pretrained exclusively on CT scans.

2. Impact of Large-Scale Pretraining on Generalization: The diverse range of CT scans within ATLAS-8K allows the model to de-

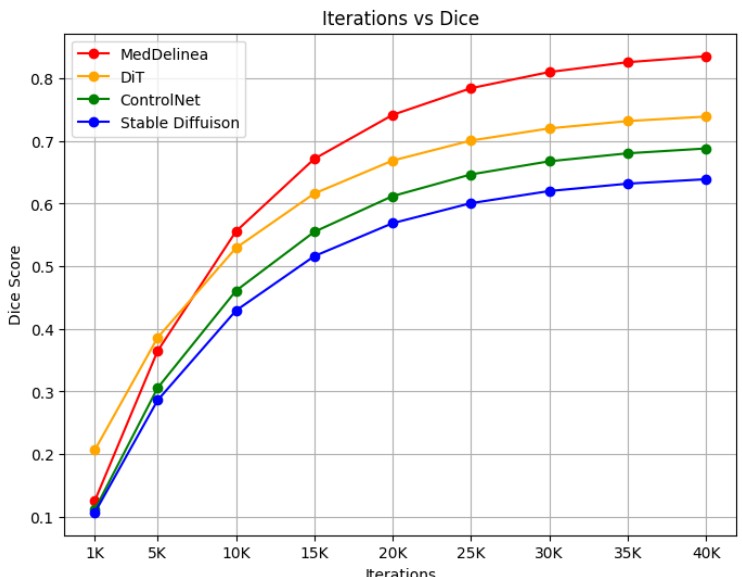

**Figure 5:** Validation Dice score progression across training iterations.

velop robust representations of organ morphology, which can be effectively leveraged for MRI segmentation. However, differences in contrast, intensity profiles, and noise distributions between CT and MRI still result in an inherent performance gap, as reflected in the lower mDSC and higher mHD95 for MRI.

Additionally, to complement the training loss curves presented in Figure 1 of the main manuscript, we provide an evaluation of model performance on the validation set using the Dice Similarity Coefficient (DSC). The purpose of this analysis is to assess the segmentation accuracy of different models throughout training and compare their convergence behavior in terms of validation performance. Figure 5 illustrates the mean Dice score computed on the validation set at different

**Table 8:** Comparison of zero-shot performance on MRI and CT data.

| Models | MRI | | CT | |
|---|---|---|---|---|
| | mDSC | mHD95 | mDSC | mHD95 |
| UNet | 73.37 | 35.45 | 74.17 | 34.01 |
| TransUnet | 74.89 | 34.47 | 75.09 | 33.65 |
| SynergyNet | 76.22 | 33.39 | 77.38 | 32.88 |
| ControlNet | 77.54 | 32.46 | 79.37 | 32.05 |
| MedSegDiff | 77.28 | 31.98 | 79.54 | 30.89 |
| VQDiffusion | 78.35 | 32.16 | 80.22 | 30.19 |
| DiT | 78.39 | 32.44 | 80.78 | 29.87 |
| nnUnet | 78.11 | 31.05 | 81.37 | 29.56 |
| nnFormer | 79.35 | 30.65 | 81.28 | 30.34 |
| UNITER++ | 79.36 | 29.09 | 82.28 | 28.89 |
| MedSAM | 80.11 | 28.98 | 82.98 | 28.21 |
| MedDelinea | **81.27** | **28.23** | **85.32** | **26.62** |

training iterations for MedDelinea and baseline models (DiT, ControlNet, and Stable Diffusion). MedDelinea consistently outperforms all baseline models across the training process, achieving the highest Dice score at convergence. On the other hand, DiT and ControlNet demonstrate steady improvement, with their Dice scores plateauing at lower values compared to MedDelinea.

## Appendix E. Qualitative Results

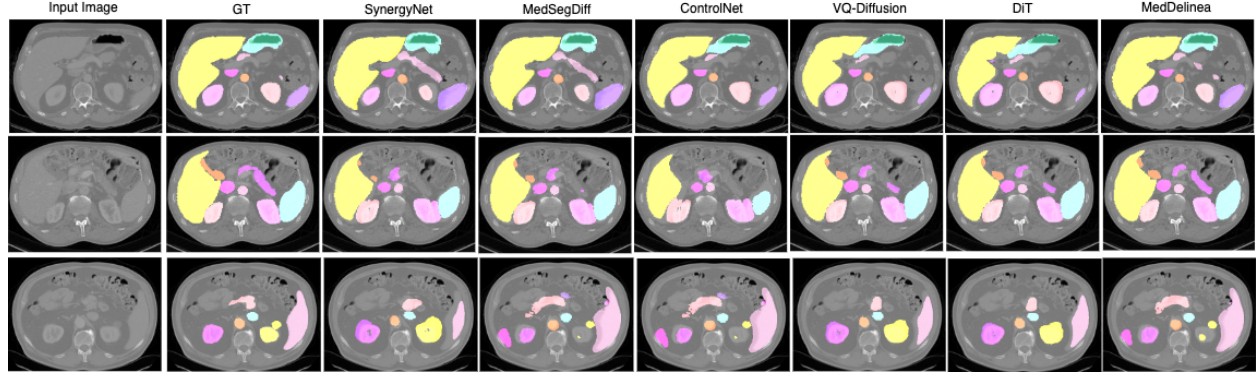

**Figure 6:** ATLAS-8k dataset results.

**Additional External Data for Test: MRI Pancreas and liver:** The qualitative results in Fig.4 highlight the segmentation performance of various models for the pancreas and liver. In the fine-tuned pancreas results (top row), **MedDelinea** closely matches the ground truth, accurately capturing the pancreas boundaries, while models like MedSegDiff and VQ-Diffusion struggle with precision. In the zero-shot pancreas results (second row), **MedDelinea** still outperforms other models, maintaining accurate segmentation without additional training, whereas DiT and MedSegDiff miss finer details. For the liver segmentation (third and fourth rows), **MedDelinea** demonstrates superior performance in both fine-tuned and zero-shot scenarios. In the fine-tuned case (third row), **MedDelinea** provides precise liver boundaries, surpassing models like SynergyNet and MedSegDiff, which miss key regions. In the zero-shot liver results (fourth row), **MedDelinea** continues to lead, while models like ControlNet and VQ-Diffusion fail to fully capture the liver's structure. Overall, **MedDelinea** excels in both settings, particularly for complex organs like the pancreas and liver, where other models struggle.

**ATLAS-8k Dataset Results:** The qualitative results in Fig. 6 show the segmentation performance of various models on the ATLAS-8k dataset (Qu et al., 2024), including SynergyNet, MedSegDiff, ControlNet, VQ-Diffusion, DiT, and **MedDelinea**. Across different input images,

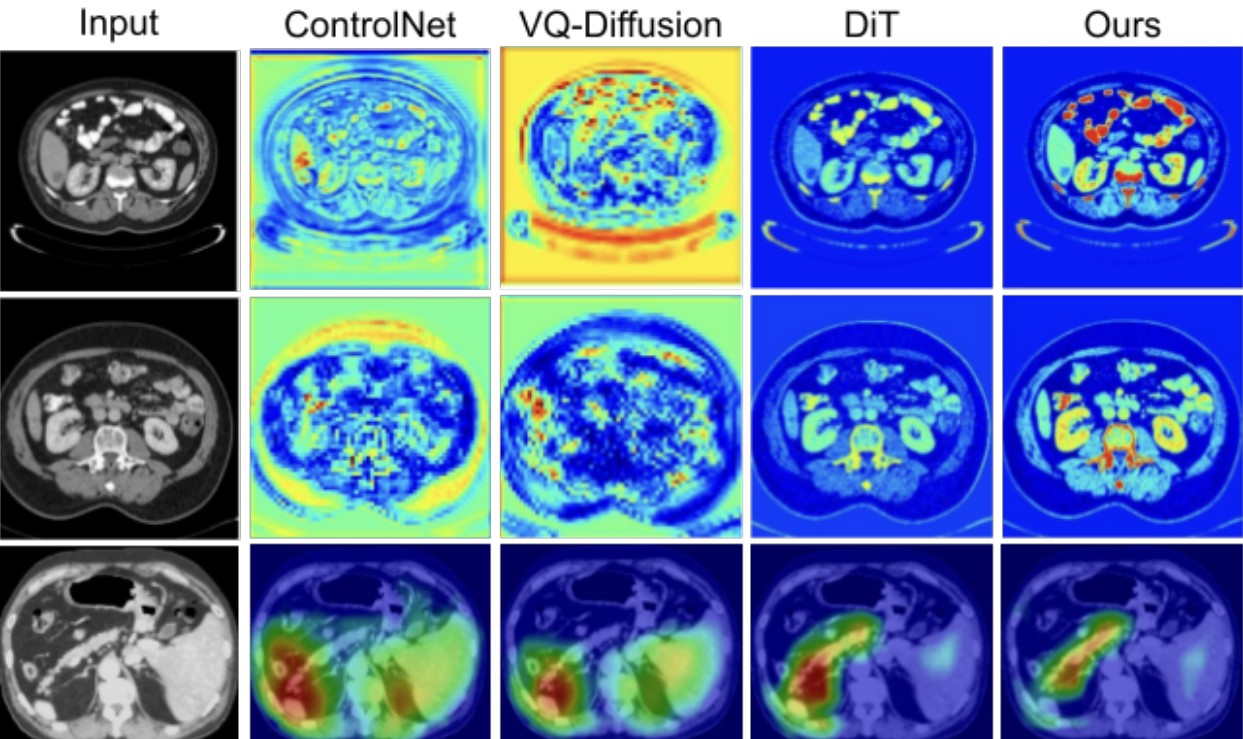

**Figure 7:** Attention maps for BTCV and AMOS datasets.

***MedDelinea*** consistently provides more accurate and precise segmentation results, closely matching the ground truth. For larger organs like the liver and spleen, ***MedDelinea*** performs well, capturing their shape and boundaries with high fidelity. Similarly, for smaller and more challenging organs, ***MedDelinea*** shows fewer segmentation errors compared to other models, which either over-segment or under-segment the regions.

**Attention Maps:** Attention maps in Fig. 7 further highlight ***MedDelinea***'s targeted attention mechanism, which focuses on fine-grained details and captures complex structures more effectively than other models. This superior attention enables ***MedDelinea*** to achieve better boundary delineation and segmentation performance, making it a robust and generalizable model for clinical applications.

## Appendix F. 3D Visualization:

In Fig. 8, shows a clearer comparison of how different models—MedSegDiff, ControlNet, DiT, and ***MedDelinea*** (ours) capture the anatomical structures from the BTCV and AMOS datasets. These visualisations help to assess the overall quality and consistency of segmentation across various organs.

***MedDelinea*** consistently produces segmentation maps that closely align with the ground truth, demonstrating its ability to capture fine details and maintain accurate boundaries, particularly in complex and small anatomical regions. In comparison, MedSegDiff and ControlNet show more inconsistencies and less precision, often struggling with challenging areas where boundary delineation is critical. DiT performs better than MedSegDiff and ControlNet but still exhibits less refinement in capturing finer structures compared to ***MedDelinea***.

For instance, in the pancreas region of the AMOS dataset, as visualized in the third row of the maps, ***MedDelinea*** produces a segmentation that closely aligns with the ground truth. The

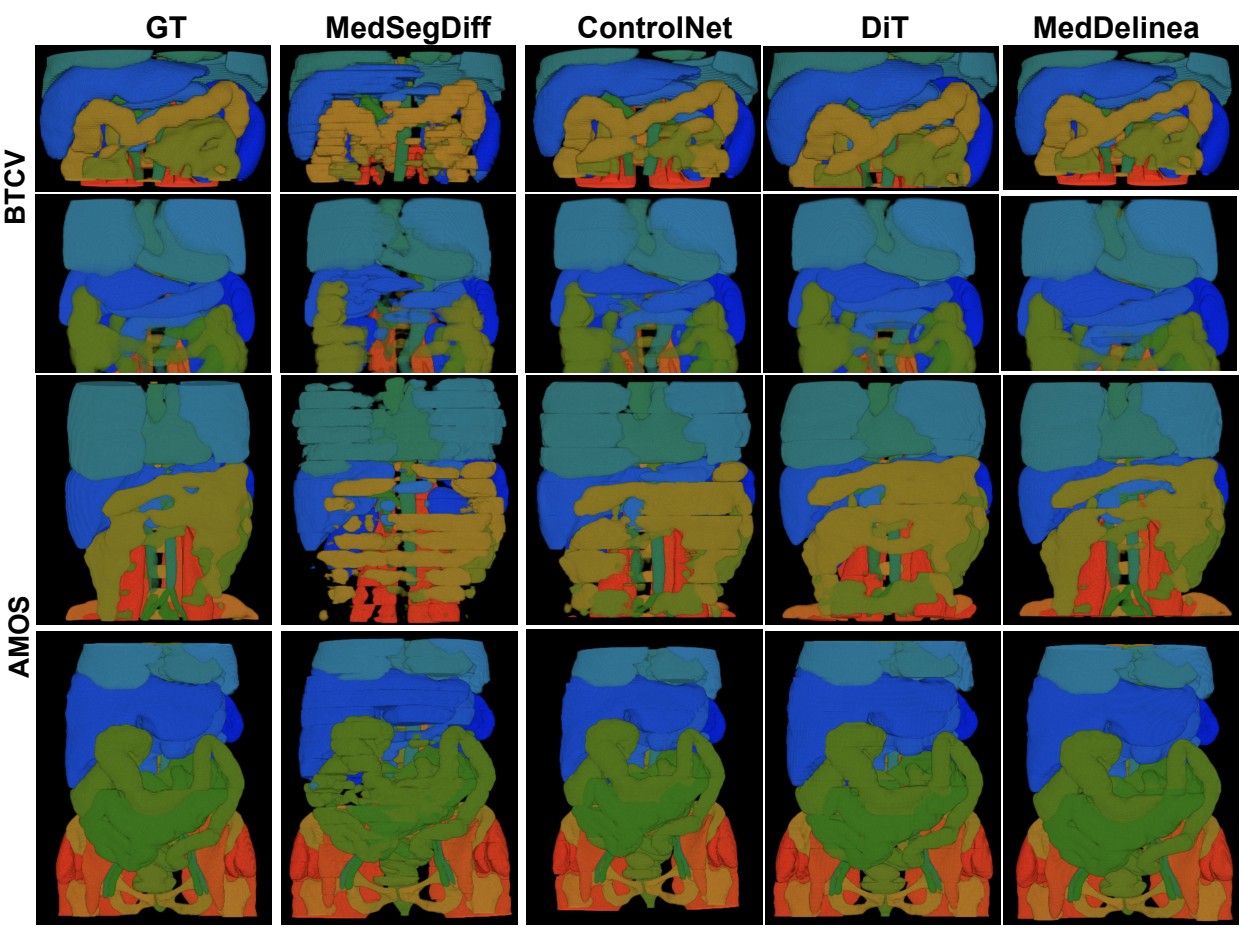

**Figure 8:** 3D Visualisation of AMOS and BTCV Dataset Segmentation Map

**Table 9:** Per organ Dice scores (%) on the ATLAS-8k dataset. Higher values indicate better segmentation performance.

| Methods | Aorta | Gallbladder | KidneyL | KidneyR | Liver | Pancreas | Spleen | Stomach |
|---|---|---|---|---|---|---|---|---|
| UNet | 67.72 | 53.37 | 64.11 | 63.34 | 86.02 | 35.67 | 77.88 | 73.21 |
| TransUnet | 74.51 | 58.92 | 67.82 | 67.54 | 87.18 | 41.34 | 81.82 | 73.98 |
| SynergyNet | 78.11 | 60.11 | 70.11 | 71.52 | 88.12 | 46.89 | 87.19 | 81.30 |
| ControlNet | 80.01 | 65.73 | 73.39 | 73.78 | 89.19 | 54.78 | 89.02 | 85.09 |
| MedSegDiff | 79.92 | 65.27 | 75.11 | 76.01 | 91.78 | 58.94 | 90.65 | 84.11 |
| VQDiffusion | 79.17 | 65.34 | 76.23 | 77.02 | 90.72 | 58.55 | 89.77 | 81.01 |
| DiT | 79.17 | 65.34 | 76.23 | 77.02 | 90.72 | 58.55 | 89.77 | 81.01 |
| *MedDelinea* | **82.34** | **68.09** | **79.01** | **80.11** | **93.14** | **61.21** | **91.72** | **86.61** |

pancreas, a smaller and more complex organ to segment, is accurately captured by ***MedDelinea*** with well-defined boundaries and minimal missing regions. In contrast, MedSegDiff and ControlNet show visible errors in this region, with fragmented or incomplete segmentations. MedSegDiff, for instance, misses part of the pancreas, leaving gaps, while ControlNet over-segments, blending boundaries with adjacent structures. DiT provides a better representation compared to MedSegDiff and ControlNet, but still lacks the sharp precision that ***MedDelinea*** demonstrates in this area.

## Appendix G. Further Related Works

**Traditional Approaches:** Medical image segmentation plays a critical role in clinical diagnostics, involving the classification of pixels in medical images (e.g., CT and MRI) to delineate anatomical structures or abnormalities. Convolutional neural network (CNN) based models, particularly UNet (Ronneberger et al., 2015) and its variants (Oktay et al., 2018; Zhang et al., 2019; Lou et al., 2021), have been widely used due to their encoder-decoder architecture, which captures both local and global context. Despite their success, CNNs struggle with long-range dependencies, limiting their effectiveness in segmenting complex anatomical structures. This has led to a shift towards transformer-based methods.

**Shift to Transformer-Based Models:** To address the limitations of CNNs, transformer-based models have been introduced. TransUNet (Chen et al., 2021) combines the strengths of CNNs for local feature extraction with transformers for long-range dependency modeling. This hybrid architecture significantly improves segmentation accuracy. Other models like Swin-UNet (Cao et al., 2022) and DS-TransUNet (Lin et al., 2022) further refine this approach by incorporating Swin Transformers, which capture multi-scale features using shifted windows and hierarchical structures. While these models enhance the understanding of complex anatomical features, they introduce challenges like increased computational complexity and reliance on large-scale pre-training datasets.

