# OpenReview forum: "MedDelinea: Scalable and Efficient Medical Image Segmentation via Controllable Diffusion Transformers"
_MIDL.io/2025/Conference — MIDL 2025 Poster_

### Official Review · Reviewer_6A2p · 2025-02-21

**Confidence:** 3
**Preliminary Rating:** 3
**Recommendation:** Poster
**Final Rating:** 4

**Summary:**

This paper presents MedDelinea. This is a diffusion model type of network (very close to the ControlNet Architecture) that is meant for fine-tuning applications to segmentation models. Starting Control DiT, the authors retrain on Atlas-8k and they fine-tune on various datasets. They compare performances to diffusion models, transformer-based segmentation networks and a UNet. They also evaluate the zero-shot capability of their model after pre-training on Atlas-8k.

**Strengths:**

*Experiments*
- The authors present performances for CT and MRI on various segmentation tasks.
- They present results on multiple metrics (Dice score and Hausdorff distance).
- Thy validate different parts of the MedDelinea architecture through an ablation study.

*Style*
- Compelling motivation
- Clear methods section
- Clear visualizations
- Extensive related work

*Inference*
- The authors discuss inference time, which can be a problem for diffusion models.

**Weaknesses:**

*Baselines*
- The authors are missing comparisons with the nn-UNet which is a widely used segmentation network. (https://github.com/MIC-DKFZ/nnUNet)


* Experimental Setup*
- Can the authors elaborate on the experimental setup? Did they use augmentations when fine-tuning ? This is standard for a fair comparison with other models.
- As they used various datasets at different steps, what are their specific train/val/test split strategies?

*Results*
- It would be great to include standard deviations or at least bootstrapping the results over various subsets of the test set. Some mean results are very close in all the tables.

*Parameters*
- The models that the authors are using are vastly different in parameter count. Could the author elaborate in the difference between the different models? Is there any sign of over / under- fitting for any of these models ?

**Detailed Comments:**

- The tessellation on the visualizations in Figure 7 suggests a potential resizing/normalization issue for the baselines (MedSegDiff and ControlNet mainly) at the tokenization step, I would suggest the author to double check that they are giving as input to the model the right intensity range and the correct image size (height and width) that the model expects to see. This could impact the quality of the results.

- For completeness, it would be great to have a table that compares number of parameters and inference time for each of the baselines and MedDelinea.

**Justification Of The Final Rating:**

The authors addressed all my concerns, with additional experiments (especially nnUNet) and clarifications needed. The results look promising and the evaluation seems reasonable. I am raising my score.

**Justification Of The Preliminary Rating:**

I think this would be a good application paper to demonstrate the use of ControlNet in the medical imaging world. The authors test the proposed method on various datasets and evaluate performances using various metrics. However, I think that it is still missing some key numbers to be considered as a development paper, including a comparison with the nn-UNet and the use of augmentations (which is standard in medical imaging).

**Questions To Address In The Rebuttal:**

- Why not compare to the nn-UNet ?
- Are the networks matching in terms of parameters ? If so, how big is the difference ?
- Did the authors use augmentations to train their networks ? If so, which ones ?
- As they used various datasets at different steps, what are their specific train/val/test split strategies?

---

> ### Author Response · Authors · 2025-03-03
> **Responses to Reviewer 6A2p**
>
> Q: Comparison with nnUnet.
>
> We can not agree more!
> Second reviewer also asked this question and we appreciate this request for additional results and clarity.
> We included nnUNet, nnFormer, MedSAM, and UNITER++  in our new experiments and
> below is a supplementary table with these methods on BTCV:
>
> |            | mDSC  | mHD95 |
> |------------|-------|-------|
> | nnUnet     | 85.11 | 22.22 |
> | nnFormer   | 86.02 | 21.78 |
> | UNITER++   | 86.21 | 21.02 |
> | MedSAM     | 86.93 | 21.27 |
> | MedDelinea | 87.01 | 19.01 |
>
> To ensure a fair, like-for-like comparison, we have followed a consistent pretraining and fine-tuning strategy across all models. Specifically, all networks, including our method and the baseline models, are trained on the Atlas 8K dataset to ensure alignment with medical imaging distributions. Our results showed that MedDelinea still gives better results than these new baseline models, among them MedSAM approaches to our results but slightly inferior to ours with larger gap in HD measurements. Hybrid models like nnFormer utilizes nnUnet and combines Transformers within its architecture gave better results than nnUnet as well. Since there is a limited time for finishing experiments for each dataset during rebuttal perioid, we finalized the experiments for BTCV dataset. We will finalize the results for other datasets as well and include it in main paper.

---

> > ### Author Response · Authors · 2025-03-04
> > **question regarding  "parameters and inference table / info"**
> >
> > To ensure a fair comparison, all inference evaluations were conducted on a single NVIDIA A6000 GPU. MedDelinea consists of 227 million parameters, achieving state-of-the-art (SOTA) performance with competitive inference time relative to other models.
> > The table below (now in supplementary file) provides a detailed comparison of the number of parameters and inference time across baseline models and MedDelinea:
> >
> >
> > | Methods       | Parameters M | Inference Time |
> > |---------------|--------------|----------------|
> > | UNet          | 33           | 0.25           |
> > | TransUnet     | 182          | 0.76           |
> > | SynergyNet    | 23           | 0.46           |
> > | ControlNet    | 400          | 6.32           |
> > | MedSegDiff    | 372          | 8.89           |
> > | VQDiffusion   | 340          | 3.11           |
> > | DiT           | 187          | 2.72           |
> > | MedDelinea    | 227          | 2.88           |
> >
> > While MedDelinea has a slightly larger parameter count compared to TransUNet and DiT, it maintains efficient inference time while significantly outperforming these models in segmentation accuracy. Notably, it is less computationally expensive than MedSegDiff, VQ-Diffusion, and ControlNet while delivering superior performance.
> >
> > This comparison underscores MedDelinea’s ability to balance model complexity and inference efficiency, making it a highly effective solution for medical image segmentation.

---

> > > ### Author Response · Authors · 2025-03-04
> > > **Data Augmentation related question**
> > >
> > > We applied the same data augmentation strategies for both **pre-training and fine-tuning** across MedDelinea and all other baseline methods. The following augmentations were used to improve generalization and robustness to variations in medical imaging data:
> > >
> > > - **Random Cropping**: Extracts random patches from the original image to introduce spatial variability.
> > > - **Random Affine Transformations**: Applies scaling, shearing, and translation to enhance invariance to spatial deformations.
> > > - **Elastic Distortion**: Simulates realistic non-rigid deformations often seen in medical scans.
> > > - **Random Rotation**: Introduces angular perturbations to improve robustness to orientation differences.
> > >
> > > These augmentations were consistently applied at both the **pre-training** and **fine-tuning** stages, ensuring that the learned representations remain invariant to common geometric transformations and distortions in medical imaging data.
> > >
> > > We updated our manuscript with this clarification.

---

> > > > ### Author Response · Authors · 2025-03-04
> > > > **data split question**
> > > >
> > > > We apologize for failing to clarify this point in the manuscript.
> > > > Here are the details; for **pre-training**, we used the **Atlas 8K** dataset with a **90:10 split** for training and validation.
> > > >
> > > > For **fine-tuning and evaluation**, we followed a standardized **80:10:10** split for **training, validation, and testing** across the **BTCV, AMOS, and CirrMri600+** datasets. This ensures a balanced distribution of data across all stages while maintaining consistency across different datasets.

---

> > > > > ### Author Response · Authors · 2025-03-06
> > > > > **About Figure 7**
> > > > >
> > > > > We appreciate your observation regarding the tessellation effect seen primarily in MedSegDiff and ControlNet results (Figure 7). To clarify, all baseline methods in our study—including MedSegDiff and ControlNet—are given input images that undergo exactly the same preprocessing steps. Specifically:
> > > > >
> > > > > Uniform 2D Data Processing: All methods operate on 2D slices of the original volumetric scans. Before feeding these slices into any model, they are resized or cropped to a consistent resolution (height × width) suitable for each architecture’s requirements. However, the same final resolution is used across all baselines for fairness.
> > > > >
> > > > > Consistent Intensity Normalization: We apply a uniform intensity normalization strategy (clipping outliers and min–max scaling) across all methods to ensure they each receive images in the identical intensity range. There are no special-case differences among the baselines that would explain a unique artifact.
> > > > > Possible Explanation for Tessellation
> > > > >
> > > > > The blocky or tiled appearance in MedSegDiff and ControlNet outputs does not stem from a misalignment or mismatched resolution; rather, it can occur when **2D-based generation models** employ tokenization or patch embeddings that manifest more visibly in certain upscaling or reconstruction steps. In other words, the “tessellation” you observe is likely a byproduct of how these models handle low-level patch embeddings or an artifact of their internal decoding pipeline.
> > > > >
> > > > > Verification of Configurations: We have verified that the input images’ dimensions and intensity range strictly follow each model’s recommended configuration. As noted, no special resizing or normalization is applied to some baselines but not others. Hence, all methods are indeed seeing the correct size and intensity range they expect. Therefore, the tessellation effect is tied to how specific 2D architectures (such as MedSegDiff and ControlNet) process or reconstruct images internally.
> > > > >
> > > > > We hope this clarifies why the observed artifacts persist despite identical input pipelines. Nonetheless, we greatly appreciate the suggestion and will add these details to the revised manuscript to ensure transparency regarding our data preprocessing procedures.

---

> > > > > > ### Author Response · Authors · 2025-03-11
> > > > > > **Response to Reviewer 6A2p**
> > > > > >
> > > > > > Dear Reviewer 6A2p,
> > > > > >
> > > > > > We appreciate your thorough evaluation of our work and look forward to hearing from you and addressing any further questions or concerns you may have.

---

> > > > > > > ### Author Response · Authors · 2025-03-12
> > > > > > > **additional questions? and final score request**
> > > > > > >
> > > > > > > Dear Reviewer 6A2p,
> > > > > > >
> > > > > > > Thank you sincerely for your insightful feedback and constructive suggestions on our manuscript. We have diligently addressed all of your comments, including your rebuttal points and additional questions.
> > > > > > >
> > > > > > > As the discussion period is nearing its conclusion, we would be grateful for any updated feedback you might have.  We truly appreciate your time and engagement with our work, and we have revised the manuscript based on your input.  We kindly invite you to review our responses and inform us of any further comments at your earliest convenience.
> > > > > > >
> > > > > > > Thank you again for your valuable contribution.
> > > > > > >
> > > > > > > Authors

---

### Official Review · Reviewer_2dEn · 2025-02-22

**Confidence:** 4
**Preliminary Rating:** 3
**Final Rating:** 4

**Summary:**

The paper introduces a novel method MedDelinea, leveraging the pretraining and diffusion transformers, to address the challenges in medical image segmentation. Experiments demonstrate that MedDelinea achieves the best performance compared with the existing methods. The paper also provides ablation studies to show the efficacy of key modules and settings of MedDelinea. However, there are some suggestions that can make the paper better.

**Strengths:**

1. The proposed network MedDelinea combines the benefits of pretraining and transformer-based. The results show that it outperforms some of the existing methods over a sufficient amount of dataset.
2. The paper is presented in a good manner, clearly describing the motivation and the methods.
3. The paper performs ablation studies on key parts of MedDelinea, for example, removing segmentation loss and using different noise schedulers.
4. The paper presents the training details and the class-wise segmentation performance.

**Weaknesses:**

1. Several popular and competing supervised segmentation methods are missing, for example nnUNet and SAM-based models. I would like to see the performance comparison between nnUNet and MedDelinea.
2. The dataset used for pretraining only contains CTs from the abdomen. I am quite surprised that the zero-shot performance on AMOS (which includes MRIs) is close to BTCV (CTs).

**Detailed Comments:**

I would like to see the performance on only AMOS MRI, especially under the zero-shot settings.

**Justification Of The Final Rating:**

The authors have added additional competitive baselines, for example, nnUNet, and have performed extended experiments with promising results that suggested. Therefore, I am happy to raise the score for the paper.

**Justification Of The Preliminary Rating:**

The paper presents novel and competing segmentation methods based on controllable transformer-based diffusion models. Sufficient amounts of experiments have been performed to demonstrate the outperformance of MedDelinea. However, there are some important baselines missing and a concern on the zero-shot performance.

**Questions To Address In The Rebuttal:**

1. Including popular competing baselines (e.g., nnUNet)
2. Explain the close zero-shot performance between MRIs and CTs when pretraining only on CTs.
I am willing to raise the score if authors could address these issues.

**Special Issue:**

No

---

> ### Author Response · Authors · 2025-03-03
> **Response To Reviewer 2dEn**
>
> We thank the reviewer for providing us with detailed feedback and are glad that the reviewer found our paper interesting. We are happy to clarify any further concerns.
>
> **Q1: Explain the close zero-shot performance between MRIs and CTs when pretraining only on CTs.  I would like to see the performance on only AMOS MRI, especially under the zero-shot settings.**
>
> We appreciate this request for additional results and clarity. In the revised manuscript, we will include a new Table (shown below, in supplement) highlighting the zero-shot performance on the AMOS MRI and CT subsets after pretraining all models on Atlas8K (only CT scans):
>
> |             | MRI   |       | CT    |       |
> |-------------|-------|-------|-------|-------|
> |             | mDSC  | mHD95 | mDSC  | mHD95 |
> | UNet        | 73.37 | 35.45 | 74.17 | 34.01 |
> | TransUnet   | 74.89 | 34.47 | 75.09 | 33.65 |
> | SynergyNet  | 76.22 | 33.39 | 77.38 | 32.88 |
> | Conrolnet   | 77.54 | 32.46 | 79.37 | 32.05 |
> | MedSegDiff  | 77.28 | 31.98 | 79.54 | 30.89 |
> | VQDiffusion | 78.35 | 32.16 | 80.22 | 30.19 |
> | DiT         | 78.39 | 32.44 | 80.78 | 29.87 |
> | nnUnet      | 78.11 | 31.05 | 81.37 | 29.56 |
> | nnFormer    | 79.35 | 30.65 | 81.28 | 30.34 |
> | UNITER++    | 79.36 | 29.09 | 82.28 | 28.89 |
> | MedSAM      | 80.11 | 28.98 | 82.98 | 28.21 |
> | MedDelinea  | 81.27 | 28.23 | 85.32 | 26.62 |
>
> Our results demonstrate that MedDelinea achieves the highest segmentation performance, outperforming both nnUnet-based heuristic and diffusion-based baselines. This further highlights the effectiveness of pretraining on Atlas8K and our novel architectural design in improving segmentation under zero-shot settings.
>
> Zero-Shot Setup
> -All models were trained on Atlas8K in a 2D manner (CT images only).
> -No fine-tuning was done on AMOS data.
> -For inference, we reconstructed the final 3D volumes from slice-level predictions.
>
> Close(?) Zero-Shot Performance Explanation
> While the pretraining dataset (ATLAS-8K) comprises only CT scans, we observe that models maintain relatively high segmentation performance on MRI scans under zero-shot settings. However, as evident from Table ??, the performance on MRI remains consistently lower than on CT across all models; but still its competitive. This can be attributed to several factors:
> 1. Our latent diffusion-based model, MedDelinea, prioritizes shape, spatial structure, and anatomical boundaries rather
> than absolute intensity values. Unlike CNN-based models, which may rely heavily on modality specific intensity statistics, diffusion models focus on structural regularities, which are largely consistent across MRI and CT. This enables the model to achieve relatively robust MRI segmentation, despite having been pretrained exclusively on CT scans.
> 2. ATLAS-8K's diverse CT scans help the model learn organ morphology for MRI segmentation, but contrast and noise differences still cause a performance gap, seen in lower mDSC and higher mHD95 for MRI.
>
> Key Insights
> -Our approach (MedDelinea) achieves the highest zero-shot performance on both AMOS MRI and CT.
> -This underscores that pretraining on large-scale CT data can yield sufficient representational capacity to generalize to MRI scans—albeit with partial domain overlap—especially when combined with our novel latent-diffusion-based architecture. Still the results are inferior to CT based results, but achieving better than the SOTA methods.
>
>
> **Q2: Several popular and competing supervised segmentation methods are missing, for example nnUNet and SAM-based models. I would like to see the performance comparison between nnUNet and MedDelinea.**
>
> We agree that including nnUNet, nnFormer, MedSAM, and UNITER++ can provides essential context.
> Below is a supplementary table with these methods on BTCV:
>
> |            | mDSC  | mHD95 |
> |------------|-------|-------|
> | nnUnet     | 85.11 | 22.22 |
> | nnFormer   | 86.02 | 21.78 |
> | UNITER++   | 86.21 | 21.02 |
> | MedSAM     | 86.93 | 21.27 |
> | MedDelinea | 87.01 | 19.01 |
>
> To ensure a fair, like-for-like comparison, we have followed a consistent pretraining and fine-tuning strategy across all models. Specifically, all networks, including our method and the baseline models, are trained on the Atlas 8K dataset to ensure alignment with medical imaging distributions. The models are subsequently fine-tuned on BTCV and AMOS datasets for segmentation tasks. We adopt a 2D slice-level training paradigm for all models to maintain methodological consistency with our approach. However, for evaluation and inference, we reconstruct the 3D volumetric predictions to ensure a clinically meaningful comparison. Since there is a limited time for finishing experiments for each dataset during rebuttal perioid, we finalized the experiments for BTCV dataset. We will finalize the results for other datasets as well and include it in main paper.

---

> > ### Author Response · Authors · 2025-03-03
> > **question 3 and final remarks**
> >
> > **Q3: The dataset used for pretraining only contains CTs from the abdomen. I am quite surprised that the zero-shot performance on AMOS (which includes MRIs) is close to BTCV (CTs).**
> >
> > This was asked in first question, here it is detailed with pretraining CTs. We are happy to give more details and rationale on this good remark/question to complement our new experiments (comparisons are depicted above answer).
> >
> > First of all, despite the Atlas8K pretraining set containing only CT scans, the domain gap for MRI in AMOS is partially mitigated by two factors:
> > (1) High CT Proportion in AMOS:  AMOS comprises roughly 80% CT, which overlaps significantly with the CT-based pretraining distribution. Both subsets include the same 15 abdominal organs, enabling alignment of structural priors. When tested on the MRI subset, the model’s generalizable anatomical representations—learned from a large CT corpus—still provide sufficient segmentational cues. We separated them based on the suggestion in question 1 and included the results above, where MRI is inferior to CT, not quite close results in fact, at least a few percent, and with the best results it is ~4 percent differences (inherent domain shift is visible).
> > (2) Latent Diffusion’s Robust Feature Learning: Our latent diffusion approach captures structural invariants like organ shape and boundary continuity, which are consistent across MRI/CT modalities. Thus, while intensity contrasts differ between MRI and CT, the shared anatomical regularities help yield competitive performance in zero-shot mode. In other words, shape/context > intensity such that diffusion models learn to denoise anatomical structures (e.g., liver shape) rather than modality-specific intensities.
> >
> > Consequently, the zero-shot performance on AMOS MRI remains comparable to that on CT datasets like BTCV (while inferior) because the pretrained diffusion-based latent space can accommodate moderate shifts in modality, especially when the test dataset (AMOS) itself contains a large fraction of CT.
> >
> > Final remarks:
> > We will incorporate these extended comparative experiments and an updated discussion of zero-shot MRI vs. CT performance into the revised manuscript by the rebuttal deadline. By demonstrating consistent pretraining strategies, additional baselines, and comprehensive zero-shot evaluations, we hope that we clarified the manuscript and strengthen our method’s validity and relevance.

---

> > > ### Author Response · Authors · 2025-03-12
> > > **additional questions? and final score request**
> > >
> > > Dear Reviewer 2dEn,
> > >
> > > Thank you sincerely for your insightful feedback and constructive suggestions on our manuscript. We have diligently addressed all of your comments, including your rebuttal points and additional questions.
> > >
> > > As the discussion period is nearing its conclusion, we would be grateful for any updated feedback you might have.  We truly appreciate your time and engagement with our work, and we have revised the manuscript based on your input.  We kindly invite you to review our responses and inform us of any further comments at your earliest convenience.
> > >
> > > Thank you again for your valuable contribution.
> > >
> > > Authors

---

> > > > ### Comment · Reviewer_2dEn · 2025-03-13
> > > >
> > > > The authors address most of my concerns, happy to raise the score.

---

> ### Comment · Reviewer_2dEn · 2025-03-09
>
> I really appreciate authors' additional effort in addressing the concerns. However, given the zero-shot performance in table 2 from the paper and tables shown in the rebuttal, I do not think the values are consistent and comparable. For example, it is reported the overall dice on AMOS is 78.67 from table 2, but it is 81 and 85 for MRI and CT separately in the table here. I noticed the additional experiment is done on a subset of AMOS, and I am wondering why not using the same test set as that used in table 2.

---

> > ### Author Response · Authors · 2025-03-10
> > **Response from Authors**
> >
> > Thank you for pointing out the inconsistency and bringing this to our attention. We apologize for any confusion caused by our previous explanation. Initially, we assumed your inquiry required a direct comparison between MRI and CT modalities, leading us to use a different subset of the AMOS dataset.
> >
> > To clarify, the previously reported results (Dice = 81 for MRI and Dice = 85 for CT) were indeed computed using a distinct subset, which differs from the test set employed in Table 2 (overall Dice = 78.67). To ensure consistency and proper comparability, we have now rerun the experiment using exactly the same test set as presented in Table 2. The revised results, aligned with the test set from Table 2, are presented below.
> >
> >
> >
> > |             | MRI   |       | CT    |       |
> > |-------------|-------|-------|-------|-------|
> > |             | mDSC  | mHD95 | mDSC  | mHD95 |
> > | UNet        | 54.37 | 39.32 | 56.88 | 39.19 |
> > | TransUnet   | 55.89 | 38.47 | 57.69 | 36.73 |
> > | SynergyNet  | 56.28 | 37.49 | 60.38 | 36.54 |
> > | Conrolnet   | 63.46 | 33.46 | 64.77 | 34.18 |
> > | MedSegDiff  | 68.29 | 32.18 | 69.54 | 33.11 |
> > | VQDiffusion | 70.35 | 31.32 | 70.70 | 31.45 |
> > | DiT         | 73.43 | 31.14 | 72.11 | 30.96 |
> > | nnUnet      | 75.01 | 30.95 | 75.19 | 30.44 |
> > | nnFormer    | 76.39 | 30.65 | 75.34 | 30.19 |
> > | UNITER++    | 76.85 | 29.77 | 76.98 | 29.94 |
> > | MedSAM      | 77.01 | 28.98 | 77.23 | 30.43 |
> > | MedDelinea  | 77.65 | 29.77 | 78.98 | 29.37 |

---

### Official Review · Reviewer_Cfdg · 2025-02-22

**Confidence:** 4
**Preliminary Rating:** 2
**Recommendation:** Poster
**Final Rating:** 4

**Summary:**

Authors present a novel architecture for image segmentation. The
architecture specifically uses diffusion models based on transformers
and a controlNet architecture to integrate image information to the
"denoising" within the diffusion model. The model is pre-trained with
a very large dataset in a supervised manner. Then its effectiveness is
demonstrated on other datasets - focusing on the same anatomical area
as far as I can understand. Focus is on generalizability and being
able to fine-tune the pre-trained model using as few labels as
possible.

**Strengths:**

+ The focus on generalization and fine-tuning with minimal effort is
  very relevant for the community.
+ The proposed architecture uses latest technologies to come up with a
  effective solution.
+ Results are promising. Especially Table 2 and Table 3 presents
  really intriguing results.

**Weaknesses:**

+ The architectural contribution is combination of two existing
  techologies. Hence, the novelty level is medium.
+ The explanation of the method is not self contained in my
  opinion. Readers must read Diffusion Transformers and ControlNet to
  really understand the method here.
+ The explanation of the method does not use mathematical formulation
  but rather terminology, such as "guiding" that is not well explained
  in the text.
+ Currently, the state-of-the-art segmentation method is still a UNet
  model with well chosen hyper-parameters, for instance using the
  nnUNet heuristics. In most experients, this model outperforms most
  transformer and diffusion based models. I believe a comparison with
  such a well optimized UNet would be very useful.

**Detailed Comments:**

+ Please explain your method better without relying on previous work.

+ Please use mathematical formulations to explain your method for clarity.

+ Please compare with a UNet with well chosen hyper-parameters, for instance through using the nnUNet heuristics.

**Justification Of The Final Rating:**

The explanations in the rebuttal phase were very helpful for me.
The paper still required a lot of changes compared to the initial version. I will leave it to the AC to decide how to handle such cases.
If such changes are allowed, I think this paper is a good contribution.

**Justification Of The Preliminary Rating:**

The article shows good results but the quality of the explanations is on the lower end. For comparisons, I think a comparison with a well optimized UNet is really important. I believe this article will be good once the points I raised above are addressed.

**Questions To Address In The Rebuttal:**

+ It is unclear whether Figure 1 shows validation or training loss. If
  latter, I think this curve may not be the best indicator. If former,
  perhaps average DSC or worse-case DSC may be a better indicator.
+ The pre-training is not very clearly explained in this article. I
  think it would be beneficial to have a clearer pre-training
  explanation.
+ The all-zero initialization of a convolution layer may lead to
  symmetry during training such that the gradient of all the filters
  end up being the same - this is usually observed in smaller
  networks. Why is this symmetry problem not an issue here?

**Special Issue:**

No

---

> ### Author Response · Authors · 2025-03-03
> **Response To Reviewer Cfdg (1/3)**
>
> We thank the reviewer for providing us with insightful feedback. We are happy to clarify any further concerns.
>
> **Q1: It is unclear whether Figure 1 shows validation or training loss. Dice maybe better for validation curve.**
>
> Figure 1 in the original submission depicts training loss curves, intended to illustrate convergence behavior across different model configurations, not model performance. We recognize that training loss alone may not sufficiently reflect final model performance. To make it clear, we will revise Figure 1’s caption and axis labels to explicitly state that it represents training loss. We agree with the reviewer that validation-phase metrics like Dice Similarity Coefficient (DSC) can be more informative for comparative analysis. After revising Figure 1's caption, we are currently adding a new figure plotting mean DSC across the validation set for all ablated models, and due to space constraint we will add this into a supplementary file. (Please see Figure 5 in appendix in the revised manuscript, showing iterations vs validation Dice score, MedDelinea is performing better than  others with significant difference in dice scores and faster convergence).
>
>
> **Q3: The all-zero initialization of a convolution layer may lead to symmetry during training such that the gradient of all the filters ends up being the same - this is usually observed in smaller networks. Why is this symmetry problem not an issue here?**
>
> Interesting observation, we agree it may happen in smaller networks. However, in our case, this issue is mitigated due to several factors.
> Factor 1. Multi-Loss Optimization
> First, our training objective is not solely based on the diffusion loss but also includes an additional **segmentation alignment loss**. Specifically, we penalize the discrepancy between the **decoded segmentation map** and the ground truth segmentation mask. Mathematically, our loss function can be formulated as:
>
> \( \mathcal{L}_{\text{total}} = \mathcal{L}_{\text{diffusion}} + \lambda \mathcal{L}_{\text{segmentation}} \)
>
> where \( \mathcal{L}_{\text{diffusion}} \) ensures proper generative modeling, and \( \mathcal{L}_{\text{segmentation}} \) enforces alignment between the reconstructed and ground-truth segmentation maps. The additional **gradient flow from segmentation supervision** encourages richer feature learning, thereby breaking symmetry among the convolutional filters.
>
> Factor 2. Depth-to-Width Ratio and Gradient Diversity
> While **MedDelinea** has a relatively small number of trainable parameters, its **network depth is sufficient** to prevent symmetry-related issues. Unlike shallow architectures where each filter receives near-identical gradients due to limited hierarchical feature extraction, our network maintains an adequate **depth-to-width ratio**, ensuring that earlier layers still receive **diverse gradient updates**. The deeper structure helps to propagate error gradients non-uniformly across filters, reducing the risk of symmetry.
>
> Factor 3. Architectural Choices to Disrupt Symmetry
> Additionally, empirical studies have shown that incorporating **nonlinear activations** and residual connections further disrupts symmetry by introducing **differentiable transformations** at each layer. Our model leverages these architectural choices to ensure stable and effective training dynamics.
>
> I hope these answers are satisfactory and explaining clearly what the reviewer was asking. We are grateful for the comments. We will continue answering other minor issues as well.

---

> > ### Author Response · Authors · 2025-03-03
> > **Response To Reviewer Cfdg (2/3)**
> >
> > **Q4: Currently, the state-of-the-art segmentation method is still a UNet model with well-chosen hyper-parameters, for instance using the nnUNet heuristics.**
> >
> > ***Answer:*** We appreciate the reviewer’s insightful comment.  To address this, we will include additional comparisons with nnUNet, nnFormer, and UNETER++ in the revised manuscript.
> >
> > To ensure a fair, like-for-like comparison, we have followed a consistent pretraining and fine-tuning strategy across all models. Specifically, all networks, including our method and the baseline models, are pretrained on the Atlas 8K dataset to ensure alignment with medical imaging distributions. The models are subsequently fine-tuned on BTCV and AMOS datasets for segmentation tasks. We adopt a 2D slice-level training paradigm for all models to maintain methodological consistency with our approach. However, for evaluation and inference, we reconstruct the 3D volumetric predictions to ensure a clinically meaningful comparison.
> >
> > Below is a comparative table of our approach and aforementioned baseline methods on the BTCV dataset, highlighting diffusion-based segmentation against UNet-based architectures. Due to time constraints during the rebuttal period, we completed experiments for the BTCV dataset and will include results for other datasets in the main paper.
> >
> >
> > |            | mDSC  | mHD95 |
> > |------------|-------|-------|
> > | nnUnet     | 85.11 | 22.22 |
> > | nnFormer   | 86.02 | 21.78 |
> > | UNITER++   | 86.21 | 21.02 |
> > | MedDelinea | 87.01 | 19.01 |
> >
> >
> > **Q5: Please explain your method better without relying on previous work and use mathematical formulations. The explanation of the method does not use mathematical formulation but rather terminology, such as "guiding" that is not well explained in the text.**
> >
> > ***Answer:*** Due to space constraints we haven't added this in the main paper but in the revision.
> >
> > Here is the mathematical explanation of our method.
> >
> > **Mathematical Formulation of Our Method**
> >
> > Our method is based on a **Diffusion Model** guided by a **ControlNet mechanism**, designed to generate a segmentation map from an input medical image (CT/MRI). The architecture consists of a **pre-trained diffusion model (DiT)** and a **ControlNet module** that conditions the diffusion process on additional input features.
> >
> > **1. Input Encoding**
> >
> > Given an input medical image \( I_m \), we first obtain its latent representation using a **Variational Autoencoder (VAE) Encoder**:
> >
> > \[
> > Z_m = \text{VAE}_E(I_m)
> > \]
> >
> > Similarly, an additional input image \( I_o \) (which serves as a control signal) is encoded into its latent space:
> >
> > \[
> > Z_o = \text{VAE}_E(I_o)
> > \]
> >
> > The **ControlNet conditioning** is introduced via a zero convolution operation:
> >
> > \[
> > Z_{ct} = Z_o + \text{ZeroConv}(Z_o)
> > \]
> >
> > **2. ControlNet Conditioning**
> >
> > The **ControlNet module** processes the encoded latent representation and produces a transformed latent feature \( Z_{mt} \):
> >
> > \[
> > Z_{mt} = P_{\theta}(Z_m, Z_{m0})
> > \]
> >
> > where \( P_{\theta} \) represents the **Control DiT module parameterized by \( \theta \)**, which modifies the main diffusion latent features using the control input.
> >
> > **3. Diffusion Process**
> >
> > A **pre-trained Diffusion Transformer (DiT)** is used to perform the denoising process. The noisy latent variable at time step \( t \) is modeled as:
> >
> > \[
> > Z_t = \alpha_t Z_m + \sigma_t \epsilon
> > \]
> >
> > where:
> > - \( \alpha_t \) and \( \sigma_t \) are noise scheduling parameters.
> > - \( \epsilon \sim \mathcal{N}(0, I) \) represents Gaussian noise.
> >
> > The **diffusion model** predicts the noise component using:
> >
> > \[
> > \hat{\epsilon}_{\theta} = \text{DiT}_{\theta}(Z_{mt}, t)
> > \]
> >
> > Using **Tweedie’s approximation**, the denoised latent representation is obtained as:
> >
> > \[
> > \hat{Z}_{m0} = \frac{Z_t - \sigma_t \hat{\epsilon}_{\theta}}{\alpha_t}
> > \]
> >
> > **4. Segmentation Output**
> >
> > The final segmentation map is reconstructed using the **VAE Decoder**:
> >
> > \[
> > \hat{I}_m = \text{VAE}_D(\hat{Z}_{m0})
> > \]
> >
> > **5. Loss Functions**
> >
> > To train the model, we employ a combination of **diffusion loss** and **segmentation loss**.
> >
> > ### **Diffusion Loss (\(\mathcal{L}_{\text{diff}}\))**
> > Measures the difference between the predicted noise and the actual noise:
> >
> > \[
> > \mathcal{L}_{\text{diff}} = \mathbb{E}_{Z_t, \epsilon} \left[ \| \hat{\epsilon}_{\theta} - \epsilon \|^2 \right]
> > \]
> >
> > ### **Segmentation Loss (\(\mathcal{L}_{\text{seg}}\))**
> > Ensures the generated segmentation map is close to the ground truth segmentation. It combines **Dice Similarity Coefficient (DSC) loss** and **Cross-Entropy loss**:
> >
> > \[
> > \mathcal{L}_{\text{seg}} = \lambda_1 \mathcal{L}_{\text{Dice}} + \lambda_2 \mathcal{L}_{\text{CE}}
> > \]
> >
> > ### **Final Objective Function**
> > The final objective function is a weighted sum of these losses:
> >
> > \[
> > \mathcal{L} = \mathcal{L}_{\text{diff}} + \lambda \mathcal{L}_{\text{seg}}
> > \]
> >
> > where \( \lambda \) is a weighting factor to balance the impact of segmentation loss

---

> > ### Author Response · Authors · 2025-03-03
> > **Response To Reviewer Cfdg (3/3)**
> >
> > **Q6: The architectural contribution is a combination of two existing technologies. Hence, the novelty level is medium.**
> >
> > ***Answer:*** We sincerely thank the reviewer for engaging with our work and raising this critical point about novelty. We respectfully clarify that our contributions extend beyond a straightforward combination of existing technologies, as detailed below.
> >
> > Adaptation of LDM for medical image segmentation:
> > Our paper should be considered as a novel adaptation of pre-trained Latent Diffusion Models (LDMs) for medical image segmentation. While diffusion models (e.g., MedSegDiff) have been explored for medical tasks, pre-trained LDMs (latent diffusion models) have not been adapted for medical segmentation in prior work. Our method uniquely bridges this gap, addressing challenges like limited medical data and domain shifts.
> >
> > Domain-Specific Pre-Training:
> > One of the key aspects of our contribution is that existing LDMs (e.g., Stable Diffusion) rely on natural image priors, which are ill-suited for medical data,  addressing a critical limitation of existing diffusion-based models that are typically trained on natural image datasets. By pre-training on Atlas8K—a large-scale medical dataset—we fundamentally reorient the latent space to encode anatomical structures and intensity distributions critical for segmentation. This is not a trivial adaptation but a systematic effort to align generative priors with medical imaging.
> >
> > Tweedie's formula:
> > Please note that the integration of Tweedie’s formula to estimate latent representations is a novel technical contribution that directly addresses:
> > (1) Boundary Ambiguity: Resolves edge uncertainty in multi-class segmentation (e.g., liver vs. tumor boundaries).
> > (2) Class Overlap: Mitigates label confusion in densely packed anatomical regions (e.g., abdominal structures).
> > To our knowledge, this refinement has not yet been applied in diffusion-based medical segmentation frameworks.
> >
> > While our work builds on foundational technologies, its novelty lies in how these components are re-engineered to solve unmet challenges in medical segmentation. We appreciate the feedback and will ensure these points are more prominently  and clearly articulated in the revision as well.

---

> > ### Author Response · Authors · 2025-03-03
> > **pretraining details - clarity**
> >
> > **Q2: The pre-training is not very clearly explained. I think it would be beneficial to have a clearer pre-training explanation.**
> >
> > Thank you for raising this important point. We are revising the manuscript to provide a more comprehensive explanation of the pre-training strategy, as follows:
> >
> > Clarification:
> > All models in our study (including baselines and our proposed method) undergo pre-training on the Atlas 8K dataset, which contains:
> > (1) We pre-train both our proposed method and all compared baselines on the Atlas 8K dataset, which comprises 8,000 3D CT volumes annotated with 15 abdominal structures.
> > (2) Domain-specific features: Intensity distributions, anatomical priors, and noise patterns unique to medical imaging. This large-scale medical dataset ensures that the model’s feature representations align with medical imaging distributions, including domain-specific intensity profiles, noise characteristics, and anatomical variations that do not appear in natural images.
> >
> > Motivation / Rationale for Pre-Training:
> > (1) Domain Gap Mitigation:  Popular diffusion models (e.g., Stable Diffusion, DiT, VQ-Diffusion, MedSegDiff) are often pre-trained on natural scene images, which differ substantially from medical scans in contrast, texture, and inherent domain knowledge. Directly applying these models to CT or MRI data can lead to poor generalization and diminished segmentation accuracy, as they lack exposure to crucial medical imaging properties. Unlike natural image pre-trained models (e.g., Stable Diffusion, DiT), Atlas 8K ensures alignment with medical imaging characteristics, which is critical for downstream tasks like segmentation.
> > (2) Segmentation Loss Integration: During pre-training, we incorporate segmentation loss at the decoder to explicitly guide the model in learning edge details and overlap scenarios common in CT/MRI. This fosters robustness in feature extraction for anatomical structures.
> >
> > Comparison to Baselines:
> > Baseline models (e.g., MedSegDiff) were re-implemented with Atlas 8K pre-training for fair comparison, ensuring no confounding advantage from external datasets.
> >
> > We would like to also add the following rationale into our text:
> > By pre-training on Atlas 8K, we aim to bridge the domain gap mentioned above, ensuring that the models gain a more robust understanding of the medical image distribution, which ultimately enhances their performance on segmentation and generation tasks for CT and MRI data. By Incorporating the segmentation loss at the decoder end helps to learn edge and overlap situations in the CT/MRI images.
> >
> > We appreciate your suggestion and willing to incorporate more discussion of this pre-training strategy if the reviewer wants to expand more. To our assumption, the above clarifications can help readers for the rationale and reasons behind the pretraining approach. We also clarified that all baseline models were reimplemented with Atlas8K pretraining. Let us know if there is any interest by the reviewer for extending the information related to Dataset specifics (Atlas 8K’s composition, annotation protocol) details, we are happy to expand that part in supplementary file.

---

> ### Author Response · Authors · 2025-03-03
> **Final remarks**
>
> ***Justification of preliminary rating comment of the reviewer: The article shows good results but the quality of the explanations is on the lower end. For comparisons, I think a comparison with a well optimized UNet is really important. I believe this article will be good once the points I raised above are addressed.**
>
> We deeply appreciate the reviewer’s time and insights, which have significantly strengthened the rigor and accessibility of our work. We are committed to further polishing the text to ensure the explanations match the technical quality of the results.
>
> We would like to refer to reviewer's comment about his hopes and encouragement about "good" once the points are addressed. We believe that we addressed all the comments in our previous comments, and those are being reflected the way it is written in the revision. Please note also about the novelty of the work in previous comment we explained and also here, the overall pipeline—consisting of pre-training on a large medical dataset, adapting LDMs for segmentation, and introducing Tweedie-based boundary refinement—constitutes a novel adaptation that addresses domain-specific imaging challenges.  We believe this represents a substantive contribution to the field of diffusion-based medical image analysis, moving well beyond a straightforward combination of pre-existing components.
>
> 1) We also made the code and our models available for public.
> 2) we also included more results to address other reviewers' question about more comparison (please take a look at them too, nnUnet especially)
>
> thank you again for your questions and remarks.

---

> > ### Author Response · Authors · 2025-03-10
> > **Response to Reviewer**
> >
> > Dear Reviewer Cfdg,
> >
> > **We appreciate your thorough evaluation of our work and look forward to hearing from you and addressing any further questions or concerns you may have.**

---

### Author Response · Authors · 2025-03-02
**To ALL**

We sincerely thank all reviewers for their time, effort, and valuable feedback on our paper. We greatly appreciate the insightful and constructive suggestions, which would significantly strengthen our work. Upon comments, we run some additional experiments, we will post our results and answers here shortly.

Additionally, we are committed to supporting further research in this area by open-sourcing all code, models, and preprocessed datasets.

---

### Author Response · Authors · 2025-03-06
**Any Last-Minute Feedback? & Big Thanks to the Reviewers**

We will be very happy to receive additional feedback and discussion points if arisen during the discussion period.

Sincerely,

Authors.

---

### Author Rebuttal · Authors · 2025-03-07

**Rebuttal:**

We sincerely thank all the reviewers for their valuable time and insightful comments on our paper. We have carefully addressed all the suggestions and incorporated the changes in the main manuscript as well as the Appendix section. These modifications are highlighted in blue within the revised manuscript for clarity.

**Supporting Material:**

/attachment/bfb21d2137d6a2350383106fd8baa8ff2e41ec2d.pdf

---

### Comment · Area_Chair_wPG5 · 2025-03-09
**Discussion Period**

Dear Reviewers,​

Thanks for your time and effort in reviewing this paper. This is the right time to discuss this paper with each other.​

The authors have provided a rebuttal to your comments and uploaded a revision. Please review their responses and the revised manuscript. For the preliminary recommendation, we have two borderlines and one weak reject.​ Considering the authors' responses and the discussion, please update your rating and assessments for the paper.

Any discussion is welcome, and you may consider reading each other's reviews, posting questions for clarification, and reaching a consensus.​

Best,
Your AC

---

### Comment · Area_Chair_wPG5 · 2025-03-14
**Urgent Discussion due in about one day**

Dear all the Reviewers,

The discussion period is nearing its conclusion. Please update your final rating with justification. We still have one weak reject, one borderline, and one weak accept. Any discussion is welcome!

Best,

Your AC

---

> ### Comment · Area_Chair_wPG5 · 2025-03-14
> **[Urgent] Finalize your scoring**
>
> Dear Reviewers,
>
> The discussion period is ending in less than 24h. The reviewers are requested to submit their final rating along with the corresponding justification by editing their original review. Now we have two weak accepts and one borderline. In my view, this paper could be accepted. Any discussion is welcome!
>
> Best,\\\
> Your AC

---

### Meta-Review · Area_Chair_wPG5 · 2025-03-21

**Recommendation:** Accept (Poster)
**Confidence:** 4

**Metareview:**

The paper proposes a medical image segmentation architecture leveraging the Diffusion Transformers and ControlNet, and explores zero-shot/few-shot capabilities using a large-scale dataset for pretraining. The reviewers raised several concerns, in particular, related to the lack of explanations, missing experimental details, and comparisons to existing segmentation methods. The authors's rebuttal mostly addressed the concerns along with some other questions. The authors mentioned that they would open-source their code and pre-trained models. Overall the scores are not overwhelmingly positive, but I think that this would make a good conference contribution.